# Learning Optimal Conformal Classifiers

**David Stutz**[1,2]**, Krishnamurthy (Dj) Dvijotham**[1]**, Ali Taylan Cemgil**[1]**, Arnaud Doucet**[1]
[1] DeepMind, [2] Max Planck Institute for Informatics, Saarland Informatics Campus

## Abstract

Modern deep learning based classifiers show very high accuracy on test data but this does *not* provide sufficient guarantees for safe deployment, especially in high-stake AI applications such as medical diagnosis. Usually, predictions are obtained without a reliable uncertainty estimate or a formal guarantee. *Conformal prediction (CP)* addresses these issues by using the classifier's predictions, *e.g.*, its probability estimates, to predict *confidence sets* containing the true class with a user-specified probability. However, using CP as a separate processing step after training prevents the underlying model from adapting to the prediction of confidence sets. Thus, this paper explores strategies to differentiate through CP *during training* with the goal of training model with the conformal wrapper *end-to-end*. In our approach, **conformal training (ConfTr)**, we specifically "simulate" conformalization on mini-batches during training. Compared to standard training, ConfTr reduces the average confidence set size (*inefficiency*) of state-of-the-art CP methods applied after training. Moreover, it allows to "shape" the confidence sets predicted at test time, which is difficult for standard CP. On experiments with several datasets, we show ConfTr can influence how inefficiency is distributed across classes, or guide the composition of confidence sets in terms of the included classes, while retaining the guarantees offered by CP.

## 1 Introduction

In classification tasks, for input $x$, we approximate the posterior distribution over classes $y \in [K] := \{1, \dots, K\}$, denoted $\pi_y(x) \approx P(Y = y | X = x)$. Following Bayes' decision rule, the *single* class with highest posterior probability is predicted for optimizing a 0-1 classification loss. This way, deep networks $\pi_{\theta,y}(x)$ with parameters $\theta$ achieve impressive accuracy on held-out test sets. However, this does not *guarantee* safe deployment. *Conformal prediction (CP)* (Vovk et al., 2005) uses a post-training calibration step to *guarantee* a user-specified *coverage*: by allowing to predict confidence sets $C(X) \subseteq [K]$, CP guarantees the true class $Y$ to be included with confidence level $\alpha$, *i.e.* $P(Y \in C(X)) \geq 1 - \alpha$ when the calibration examples $(X_i, Y_i)$, $i \in I_{\text{cal}}$ are drawn exchangeably from the test distribution. This is usually achieved in two steps: In the *prediction step*, so-called *conformity scores* (w.r.t. to a class $k \in [K]$) are computed to construct the confidence sets $C(X)$. During the *calibration step*, these conformity scores on the calibration set w.r.t. the true class $Y_i$ are ranked to determine a cut-off threshold $\tau$ for the predicted probabilities $\pi_\theta(x)$ guaranteeing coverage $1 - \alpha$. This is called *marginal* coverage as it holds only unconditionally, *i.e.*, the expectation is being taken not only w.r.t. $(X, Y)$ but also over the distribution of all possible calibration sets, rather than w.r.t. the conditional distribution $p(Y|X)$.

CP also outputs intuitive uncertainty estimates: larger confidence sets $|C(X)|$ generally convey higher uncertainty. Although CP is agnostic to details of the underlying model $\pi_\theta(x)$, the obtained uncertainty estimates depend strongly on the model's performance. If the underlying classifier is poor, CP results in too large and thus uninformative confidence sets. "Uneven" coverage is also a common issue, where lower coverage is achieved on more difficult classes. To address such problems, the threshold CP method of (Sadinle et al., 2019) explicitly minimizes inefficiency. Romano et al. (2020) and Cauchois et al. (2020) propose methods that perform favorably in terms of (approximate) conditional coverage. The *adaptive prediction sets (APS)* method of Romano et al. (2020) is further extended by Angelopoulos et al. (2021) to return smaller confidence sets. These various objectives are typically achieved by changing the definition of the conformity scores. In all cases, CP is used as a post-training calibration step. In contrast, our work does *not* focus on advancing CP

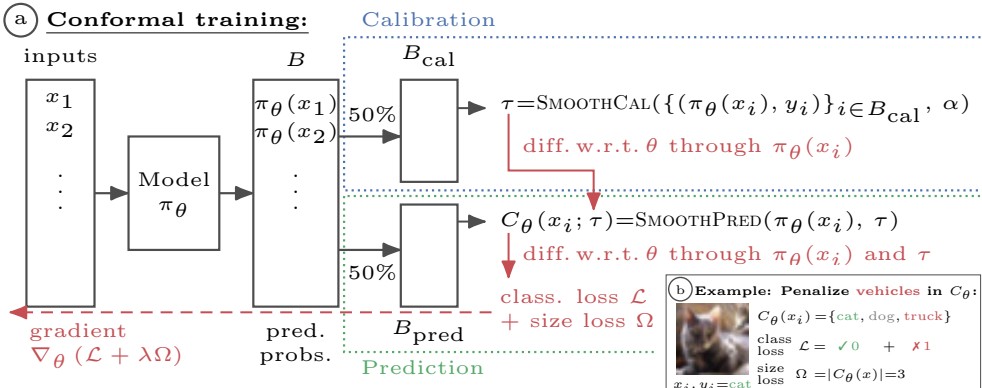

Figure 1: **Illustration of *conformal training (ConfTr)*:** We develop differentiable prediction and calibration steps for *conformal prediction (CP)*, SMOOTHCAL and SMOOTHPRED. During training, this allows ConfTr to "simulate" CP on each mini-batch $B$ by calibrating on the first half $B_{cal}$ and predicting confidence sets on the other half $B_{pred}$ (*c.f.* (a)). ConfTr can optimize arbitrary losses on the predicted confidence sets, *e.g.*, reducing average confidence set size (*inefficiency*) using a size loss $\Omega$ or penalizing specific classes from being included using a classification loss $\mathcal{L}$ (*c.f.* (b)). *After* training using our method, *any* existing CP method can be used to obtain a coverage guarantee.

itself, *e.g.*, through new conformity scores, but develops a novel training procedure for the classifier $\pi_\theta$. After training, any of the above CP methods can readily be applied.

Indeed, while the flexibility of CP regarding the underlying model appears attractive, it is also a severe limitation: Learning the model parameters $\theta$ is *not* informed about the post-hoc "conformalization", *i.e.*, they are are not tuned towards any specific objective such as reducing expected confidence set size (*inefficiency*). During training, the model will typically be trained to minimize cross-entropy loss. At test time, in contrast, it is used to obtain a set predictor $C(X)$ with specific properties such as low inefficiency. In concurrent work, Bellotti (2021) addresses this issue by learning a set predictor $C(X)$ through thresholding logits: Classes with logits exceeding 1 are included in $C(X)$ and training aims to minimize inefficiency while targeting coverage $1 - \alpha$. In experiments using linear models only, this approach is shown to decrease inefficiency. However, (Bellotti, 2021) ignores the crucial calibration step of CP during training and does *not* allow to optimize losses beyond marginal coverage or inefficiency. In contrast, our work subsumes (Bellotti, 2021), but additionally considers the calibration step during training, which is crucial for further decreasing inefficiency. Furthermore, we aim to allow fine-grained control over class-conditional inefficiency or the composition of the confidence sets by allowing to optimize arbitrary losses defined on confidence sets.

Our **contributions** can be summarized as follows:

1. We propose **conformal training (ConfTr)**, a procedure allowing to train model and conformal wrapper *end-to-end*. This is achieved by developing smooth implementations of recent CP methods for use during training. On each mini-batch, ConfTr "simulates" conformalization, using half of the batch for calibration, and the other half for prediction and loss computation, *c.f.* Fig. 1 (a). After training, *any* existing CP method can provide a coverage guarantee.

2. In experiments, using ConfTr for training consistently reduces the inefficiency of conformal predictors such as *threshold CP (*THR*)* (Sadinle et al., 2019) or APS (Romano et al., 2020) applied *after* training. We further improve over (Bellotti, 2021), illustrating the importance of the calibration step during training.

3. Using carefully constructed losses, ConfTr allows to "shape" the confidence sets obtained at test time: We can reduce *class-conditional* inefficiency or "coverage confusion", *i.e.*, the likelihood of two or more classes being included in the same confidence sets, *c.f.* Fig. 1 (b). Generally, in contrast to (Bellotti, 2021), ConfTr allows to optimize arbitrary losses on the confidence sets.

Because ConfTr is agnostic to the CP method used at test time, our work is complementary to most related work, *i.e.*, *any* advancement in terms of CP is directly applicable to ConfTr. For example, this might include conditional or application-specific guarantees as in (Sadinle et al., 2016; Bates et al., 2021). Most importantly, ConfTr preserves the coverage guarantee obtained through CP.

| CP Baseline Comparison by Ineff | | | | |
|---|---|---|---|---|
| Dataset, $\alpha$ | THRL | THR | APS | RAPS |
| CIFAR10, 0.05 | 2.22 | **1.64** | 2.06 | 1.74 |
| CIFAR10, 0.01 | 3.92 | **2.93** | 3.30 | 3.06 |
| CIFAR100, 0.01 | 19.22 | **10.63** | 16.62 | 14.25 |

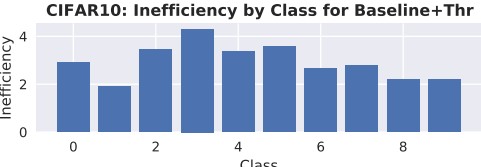

Figure 2: **Baseline CP Results on CIFAR:** *Left:* Inefficiency (Ineff, lower is better) for the CP methods discussed in Sec. 2. Coverage (Cover), omitted here, is empirically close to $1 - \alpha$. THR clearly outperforms all approaches w.r.t. inefficiency. *Right:* Inefficiency distribution across CIFAR10 classes (for $\alpha$=0.01) is plotted, with more difficult classes yielding higher inefficiency.

## 2 DIFFERENTIABLE CONFORMAL PREDICTORS

We are interested in training the model $\pi_\theta$ end-to-end with the conformal wrapper in order to allow fine-grained control over the confidence sets $C(X)$. Before developing differentiable CP methods for training in Sec. 2.2, we review two recently proposed conformal predictors that we use at test time. These consist of two steps, see Sec. 2.1: for *prediction* (on the test set) we need to define the confidence sets $C_\theta(X; \tau)$ which depend on the model parameters $\theta$ through the predictions $\pi_\theta$ and where the threshold $\tau$ is determined during *calibration* on a held-out calibration set $(X_i, Y_i), i \in I_{\text{cal}}$ in order to obtain coverage.

### 2.1 CONFORMAL PREDICTORS

The **threshold conformal predictor (THR)** (Sadinle et al., 2019) constructs the confidence sets by thresholding probabilities: $C_\theta(x; \tau) := \{k : \pi_{\theta,k}(x) =: E_\theta(x, k) \geq \tau\}$. Here, the subscript $C_\theta$ makes the dependence on the model $\pi_\theta$ and its parameters $\theta$ explicit. During calibration, $\tau$ is computed as the $\alpha(1 + 1/|I_{\text{cal}}|)$-quantile of the so-called conformity scores $E_\theta(x_i, y_i) = \pi_{\theta,y_i}(x_i)$. The conformity scores indicate, for each example, the threshold that ensures coverage. Marginal coverage of $(1 - \alpha)$ is guaranteed on a test example $(X, Y)$. In practice, THR can also be applied on logits (THRL) or log-probabilities (THRLP) instead of probabilities.

**Adaptive Prediction Sets (APS)** (Romano et al., 2020) constructs confidence sets based on the ordered probabilities. Specifically, $C_\theta(x; \tau) := \{k : E_\theta(x, k) \leq \tau\}$ with:

$$E_\theta(x, k) := \pi_{\theta,y^{(1)}}(x) + \ldots + \pi_{\theta,y^{(k-1)}}(x) + U\pi_{\theta,y^{(k)}}(x), \tag{1}$$

where $\pi_{\theta,y^{(1)}}(x) \geq \ldots \geq \pi_{\theta,y^{(K)}}(x)$ and $U$ is a uniform random variable in $[0, 1]$ to break ties. Similar to THR, the conformity scores $E_\theta(x_i, y_i)$ w.r.t. the true classes $y_i$ are used for calibration, but the $(1 - \alpha)(1 + 1/|I_{\text{cal}}|)$-quantile is required to ensure marginal coverage on test examples.

Performance of CP is then measured using two metrics: (empirical and marginal) **coverage (Cover)** as well as **inefficiency (Ineff)**. Letting $I_{\text{test}}$ be a test set of size $|I_{\text{test}}|$, these metrics are computed as

$$\text{Cover} := \frac{1}{|I_{\text{test}}|} \sum_{i \in I_{\text{test}}} \delta[y_i \in C(x_i)] \quad \text{and} \quad \text{Ineff} := \frac{1}{|I_{\text{test}}|} \sum_{i \in I_{\text{test}}} |C(x_i)|, \tag{2}$$

where $\delta$ denotes an indicator function that is 1 when its argument is true and 0 otherwise. Due to the marginal coverage guarantee provided by CP (*c.f.* (Romano et al., 2020) or App. C), the empirical coverage, when averaged across several calibration/test splits, is Cover $\approx 1 - \alpha$. Thus, we concentrate on inefficiency as the main metric to compare across CP methods and models. With *accuracy*, we refer to the (top-1) accuracy with respect to the $\arg\max$-predictions, *i.e.*, $\arg\max_k \pi_{\theta,k}(x)$, obtained by the underlying model $\pi$. As shown in Fig. 2 (left), THR clearly outperforms THRL and APS w.r.t. inefficiency (lower is better) averaged across random $I_{\text{cal}}/I_{\text{test}}$ splits (details in Sec. F).

CP is intended to be used as a "wrapper" around $\pi_\theta$. "Better" CP methods generally result in lower inefficiency for a *fixed* model $\pi_\theta$. For example, following Fig. 2 (left), regularized APS (RAPS) (Angelopoulos et al., 2021) recently showed how to improve inefficiency compared to APS by modifying the conformity score – without outperforming THR, however. Fine-grained control over inefficiency, *e.g.*, conditioned on the class or the composition of the $C(X)$ is generally *not* possible. Integrating CP into the training procedure promises a higher degree of control, however, requires differentiable CP implementations, *e.g.*, for THR or APS.

```
1: function PREDICT(π_θ(x), τ)
2:    compute E_θ(x, k), k∈[K]
3:    return C_θ(x; τ) = {k : E_θ(x, k) ≥ τ}
```

```
1: function CALIBRATE({(π_θ(x_i), y_i)}_{i=1}^n, α)
2:    compute E_θ(x_i, y_i), i=1, ..., n
3:    return QUANTILE({E_θ(x_i, y_i)}, α(1 + 1/n))
```

```
1: function SMOOTHPRED(π_θ(x), τ, T=1)
2:    return C_{θ,k}(x; τ) = σ((E_θ(x,k)−τ)/T), k ∈ [K]
3: function SMOOTHCAL({(π_θ(x_i), y_i)}_{i=1}^n, α)
4:    return SMOOTHQUANT({E_θ(x_i, y_i)}, α(1 + 1/n))
```

```
1: function CONFORMALTRAINING(α, λ=1)
2:    for mini-batch B do
3:       randomly split batch B_cal ⊎ B_pred = B
4:       {"On-the-fly" calibration on B_cal:}
5:       τ = SMOOTHCAL({(π_θ(x_i), y_i)}_{i∈B_cal}, α)
6:       {Prediction only on i ∈ B_pred:}
7:       C_θ(x_i; τ) = SMOOTHPRED(π_θ(x_i), τ)
8:       {Optional classification loss:}
9:       L_B = 0 or Σ_{i∈B_pred} L(C_θ(x_i; τ), y_i)
10:      Ω_B = Σ_{i∈B_pred} Ω(C_θ(x_i; τ))
11:      Δ = ∇_θ 1/|B_pred|(L_B + λΩ_B)
12:      update parameters θ using Δ
```

Algorithm 1: **Smooth CP and Conformal Training (ConfTr):** *Top left:* At test time, for THR, PREDICT computes the conformity scores $E_\theta(x, k)$ for each $k \in [K]$ and constructs the confidence sets $C_\theta(x; \tau)$ by thresholding with $\tau$. CALIBRATE determines the threshold $\tau$ as the $\alpha(1 + 1/n)$-quantile of the conformity scores w.r.t. the true classes $y_i$ on a calibration set $\{(x_i, y_i)\}$ of size $n := |I_{\text{cal}}|$. THR and APS use different conformity scores. *Right and bottom left:* ConfTr calibrates on a part of each mini-batch, $B_{\text{cal}}$. Thereby, we obtain guaranteed coverage on the other part, $B_{\text{pred}}$ (in expectation across batches). Then, the inefficiency on $B_{\text{pred}}$ is minimized to update the model parameters $\theta$. Smooth implementations of calibration and prediction are used.

## 2.2    DIFFERENTIABLE PREDICTION AND CALIBRATION STEPS

Differentiating through CP involves differentiable prediction and calibration steps: We want $C_\theta(x; \tau)$ to be differentiable w.r.t. the predictions $\pi_\theta(x)$, and $\tau$ to be differentiable w.r.t. to the predictions $\pi_\theta(x_i)$, $i \in I_{\text{cal}}$ used for calibration. We emphasize that, ultimately, this allows to differentiate through both calibration and prediction w.r.t. the *model parameters* $\theta$, on which the predictions $\pi_\theta(x)$ and thus the conformity scores $E_\theta(x, k)$ depend. For brevity, we focus on THR, see Alg. 1 and discuss APS in App. D.

**Prediction** involves thresholding the conformity scores $E_\theta(x, k)$, which can be smoothed using the sigmoid function $\sigma(z) = 1/1+\exp(−z)$ and a temperature hyper-parameter $T$: $C_{\theta,k}(x; \tau) := \sigma((E_\theta(x,k)−\tau)/T)$. Essentially, $C_{\theta,k}(x; \tau) \in [0, 1]$ represents a *soft* assignment of class $k$ to the confidence set, *i.e.*, can be interpreted as the probability of $k$ being included. For $T \to 0$, the "hard" confidence set will be recovered, *i.e.*, $C_{\theta,k}(x; \tau) = 1$ for $k \in C_\theta(x; \tau)$ and 0 otherwise. For THR, the conformity scores are naturally differentiable w.r.t. to the parameters $\theta$ because $E(x, k) = \pi_{\theta,k}(x)$.

As the conformity scores are already differentiable, **calibration** merely involves a differentiable quantile computation. This can be accomplished using any smooth sorting approach (Blondel et al., 2020; Cuturi et al., 2019; Williamson, 2020). These often come with a "dispersion" hyper-parameter $\epsilon$ such that smooth sorting approximates "hard" sorting for $\epsilon \to 0$. Overall, this results in the threshold $\tau$ being differentiable w.r.t. the predictions of the calibration examples $\{(\pi_\theta(x_i), y_i)\}_{i \in I_{\text{cal}}}$ and the model's parameters $\theta$.

As this approximation is using smooth operations, the coverage guarantee seems lost. However, in the limit of $T, \epsilon \to 0$ we recover the original non-smooth computations and the corresponding coverage guarantee. Thus, it is reasonable to assume that, in practice, we *empirically* obtain coverage close to $(1 − \alpha)$. We found that this is sufficient because these smooth variants are *only* used during training. At test time, we use the original (non-smooth) implementations and the coverage guarantee follows directly from (Romano et al., 2020; Sadinle et al., 2019).

## 3    CONFORMAL TRAINING (CONFTR): *Learning* CONFORMAL PREDICTION

The key idea of **conformal training (ConfTr)** is to "simulate" CP during training, *i.e.*, performing both calibration and prediction steps on each mini-batch. This is accomplished using the differentiable conformal predictors as introduced in Sec. 2.2. ConfTr can be viewed as a generalization of (Bellotti, 2021) that just differentiates through the prediction step with a fixed threshold, without considering the crucial calibration step, see App. E. In both cases, *only* the training procedure changes. After training, standard (non-smooth) conformal predictors are applied.

### 3.1 CONFTR BY OPTIMIZING INEFFICIENCY

ConfTr performs (differentiable) CP on each mini-batch during stochastic gradient descent (SGD) training. In particular, as illustrated in Fig. 1 ⓐ, we split each mini-batch $B$ in half: the first half is used for calibration, $B_{\text{cal}}$, and the second one for prediction and loss computation, $B_{\text{pred}}$. That is, on $B_{\text{cal}}$, we calibrate $\tau$ by computing the $\alpha(1 + {}^1/_{|B_{\text{cal}}|})$-quantile of the conformity scores in a differentiable manner. It is important to note that we compute $C_\theta(x_i; \tau)$ only for $i \in B_{\text{pred}}$ and *not* for $i \in B_{\text{cal}}$. Then, in expectation across mini-batches and large enough $|B_{\text{cal}}|$, for $T, \epsilon \to 0$, CP guarantees coverage $1 - \alpha$ on $B_{\text{pred}}$. Assuming empirical coverage to be close to $(1 - \alpha)$ in practice, we only need to minimize inefficiency during training:

$$\min_\theta \log \mathbb{E}\left[\Omega(C_\theta(X; \tau))\right] \quad \text{with } \Omega(C_\theta(x; \tau)) = \max\left(0, \sum_{k=1}^K C_{\theta,k}(x; \tau) - \kappa\right). \tag{3}$$

We emphasize that ConfTr optimizes the model parameters $\theta$ on which the confidence sets $C_\theta$ depend through the model predictions $\pi_\theta$. Here, $\Omega$ is a "smooth" *size loss* intended to minimize the expected inefficiency, *i.e.*, $\mathbb{E}[|C_\theta(X; \tau)|]$, not to be confused with the statistic in Eq. (2) used for evaluation. Remember that $C_{\pi,k}(x; \tau)$ can be understood as a soft assignment of class $k$ to the confidence set $C_\theta(x; \tau)$. By default, we use $\kappa = 1$ in order to not penalize singletons. However, $\kappa \in \{0, 1\}$ can generally be treated as hyper-parameter. After training, any CP method can be applied to re-calibrate $\tau$ on a held-out calibration set $I_{\text{cal}}$ as usual, *i.e.*, the thresholds $\tau$ obtained during training are *not* kept. This ensures that we obtain a coverage guarantee of CP.

### 3.2 CONFTR WITH CLASSIFICATION LOSS

In order to obtain more control over the composition of confidence sets $C_\theta(X; \tau)$ at test time, ConfTr can be complemented using a generic loss $\mathcal{L}$:

$$\min_\theta \log\left(\mathbb{E}\left[\mathcal{L}(C_\theta(X; \tau), Y) + \lambda\Omega(C_\theta(X; \tau))\right]\right). \tag{4}$$

While $\mathcal{L}$ can be any arbitrary loss defined directly on the confidence sets $C_\theta$, we propose to use a "configurable" *classification loss* $\mathcal{L}_{\text{class}}$. This classification loss is intended to explicitly enforce coverage, *i.e.*, make sure the true label $Y$ is included in $C_\theta(X; \tau)$, and optionally penalize other classes $k$ *not* to be included in $C_\theta$, as illustrated in Fig. 1 ⓑ. To this end, we define

$$\mathcal{L}_{\text{class}}(C_\theta(x; \tau), y) := \sum_{k=1}^K L_{y,k}\bigg[\underbrace{(1 - C_{\theta,k}(x; \tau)) \cdot \delta[y = k]}_{\text{enforce } y \text{ to be in } C} + \underbrace{C_{\theta,k}(x; \tau) \cdot \delta[y \neq k]}_{\text{penalize class } k \neq y \text{ not to be in } C}\bigg]. \tag{5}$$

As above, $C_{\theta,k}(x; \tau) \in [0, 1]$ such that $1 - C_\theta(x; \tau)$ can be understood as the likelihood of $k$ not being in $C_\theta(x; \tau)$. In Eq. (5), the first term is used to encourage coverage, while the second term can be used to avoid predicting other classes. This is governed by the *loss matrix* $L$: For $L = I_K$, *i.e.*, the identity matrix with $K$ rows and columns, this loss simply enforces coverage (perfect coverage if $\mathcal{L}_{\text{class}} = 0$). However, setting any $L_{y,k} > 0$ for $y \neq k$ penalizes the model from including class $k$ in confidence sets with ground truth $y$. Thus, cleverly defining $L$ allows to define rather complex objectives, as we will explore next. ConfTr with (optional) classification loss is summarized in Alg. 1 (right) and Python code can be found in App. P.

### 3.3 CONFTR WITH GENERAL AND APPLICATION-SPECIFIC LOSSES

We consider several use cases motivated by medical diagnosis, *e.g.*, breast cancer screening (McKinney et al., 2020) or classification of dermatological conditions (Liu et al., 2020; Roy et al., 2021; Jain et al., 2021). In skin condition classification, for example, predicting sets of classes, *e.g.*, the top-$k$ conditions, is already a common strategy for handling uncertainty. In these cases, we not only care about coverage guarantees but also desirable characteristics of the confidence sets. These constraints in terms of the predicted confidence sets can, however, be rather complicated and pose difficulties for standard CP. We explore several exemplary use cases to demonstrate the applicability of ConfTr, that are also relevant beyond the considered use cases in medical diagnosis.

First, we consider "shaping" class-conditional inefficiency, formally defined as

$$\text{Ineff}[Y = y] := \frac{1}{\sum_{i \in I_{\text{test}}} \delta[y_i = y]} \sum_{i \in I_{\text{test}}} \delta[y_i = y] |C(x_i)|. \tag{6}$$

Similarly, we can define inefficiency conditional on a *group* of classes. For example, we could reduce inefficiency, *i.e.*, uncertainty, on "low-risk" diseases at the expense of higher uncertainty on "high-risk" conditions. This can be thought of as re-allocating time spent by a doctor towards high-risk cases. Using ConfTr, we can manipulate group- or class-conditional inefficiency using a weighted size loss $\omega \cdot \Omega(C(X; \tau))$ with $\omega := \omega(Y)$ depending on the ground truth $Y$ in Eq. (3).

Next, we consider *which* classes are actually included in the confidence sets. CP itself does not enforce any constraints on the composition of the confidence sets. However, with ConfTr, we can penalize the "confusion" between pairs of classes: for example if two diseases are frequently confused by doctors, it makes sense to train models that avoid confidence sets that contain *both* diseases. To control such cases, we define the *coverage confusion matrix* as

$$\Sigma_{y,k} := \frac{1}{|I_{\text{test}}|} \sum_{i \in I_{\text{test}}} \delta[y_i = y \wedge k \in C(x_i)]. \tag{7}$$

The off-diagonals, *i.e.*, $\Sigma_{y,k}$ for $y \neq k$, quantify how often class $k$ is included in confidence sets with true class $y$. Reducing $\Sigma_{y,k}$ can be accomplished using a positive entry $L_{y,k} > 0$ in Eq. (5).

Finally, we explicitly want to penalize "overlap" between groups of classes in confidence sets. For example, we may *not* want to concurrently include very high-risk conditions among low-risk ones in confidence sets, to avoid unwanted anxiety or tests for the patient. Letting $K_0 \uplus K_1$ being two disjoint sets of classes, we define *mis-coverage* as

$$\text{MisCover}_{0 \rightarrow 1} = \frac{1}{\sum_{i \in I_{\text{test}}} \delta[y_i \in K_0]} \sum_{i \in I_{\text{test}}} \delta[y_i \in K_0 \wedge (\exists k \in K_1 : k \in C(x_i))]. \tag{8}$$

Reducing $\text{MisCover}_{0 \rightarrow 1}$ means avoiding classes $K_1$ being included in confidence sets of classes $K_0$. Again, we use $L_{y,k} > 0$ for $y \in K_0, k \in K_1$ to approach this problem. $\text{MisCover}_{1 \rightarrow 0}$ is defined analogously and measures the opposite, *i.e.*, classes $K_0$ being included in confidence sets of $K_1$.

## 4 EXPERIMENTS

We present experiments in two parts: First, in Sec. 4.1, we demonstrate that ConfTr can reduce inefficiency of THR and APS compared to CP applied to a baseline model trained using cross-entropy loss separately (see Tab. 1 for the main results). Thereby, we outperform concurrent work of Bellotti (2021). Second, in Sec. 4.2, we show how ConfTr can be used to "shape" confidence sets, *i.e.*, reduce class-conditional inefficiency for specific (groups of) classes or coverage confusion of two or more classes, while maintaining the marginal coverage guarantee. This is impossible using (Bellotti, 2021) and rather difficult for standard CP.

We consider several benchmark datasets as well as architectures, *c.f.* Tab. A, and report metrics averaged across 10 random calibration/test splits for 10 trained models for each method. We focus on (non-differentiable) THR and APS as CP methods used *after* training and, thus, obtain the corresponding coverage guarantee. THR, in particular, consistently achieves lower inefficiency for a fixed confidence level $\alpha$ than, *e.g.*, THRL (*i.e.*, THR on logits) or RAPS, see Fig. 2 (left). We set $\alpha = 0.01$ and use the same $\alpha$ during training using ConfTr. Hyper-parameters are optimized for THR or APS individually. We refer to App. F for further details on datasets, models, evaluation protocol and hyper-parameter optimization.

### 4.1 REDUCING INEFFICIENCY WITH CONFTR

In the first part, we focus on the inefficiency reductions of ConfTr in comparison to a standard cross-entropy training baseline and (Bellotti, 2021) (Bel). After summarizing the possible inefficiency reductions, we also discuss which CP method to use during training and how ConfTr can be used for ensembles and generalizes to lower $\alpha$.

Table 1: **Main Inefficiency Results**, comparing (Bellotti, 2021) (Bel, trained with THRL) and ConfTr (trained with THRLP) using THR or APS at test time (with $\alpha=0.01$). We also report improvements relative to the baseline, *i.e.*, standard cross-entropy training, in percentage in parentheses. ConfTr results in a consistent improvement of inefficiency for both THR and APS. Training with $\mathcal{L}_{\text{class}}$, using $L = I_K$, generally works slightly better. On CIFAR, the inefficiency reduction is smaller compared to other datasets as ConfTr is trained on pre-trained ResNet features, see text. More results can be found in App. J.

| Inefficiency ↓, ConfTr (trained w/ THRLP), $\alpha = 0.01$ | | | | | | | |
|---|---|---|---|---|---|---|---|
| | THR | | | | APS | | |
| Dataset | Basel. | Bel | ConfTr | $+\mathcal{L}_{\text{class}}$ | Basel. | ConfTr | $+\mathcal{L}_{\text{class}}$ |
| MNIST | 2.23 | 2.70 | 2.18 | **2.11** (-5.4%) | 2.50 | 2.16 | **2.14** (-14.4%) |
| F-MNIST | 2.05 | 1.90 | 1.69 | **1.67** (-18.5%) | 2.36 | 1.82 | **1.72** (-27.1%) |
| EMNIST | 2.66 | 3.48 | 2.66 | **2.49** (-6.4%) | 4.23 | **2.86** | 2.87 (-32.2%) |
| CIFAR10 | 2.93 | 2.93 | 2.88 | **2.84** (-3.1%) | 3.30 | 3.05 | **2.93** (-11.2%) |
| CIFAR100 | 10.63 | 10.91 | 10.78 | **10.44** (-1.8%) | 16.62 | 12.99 | **12.73** (-23.4%) |

**Main Results:** In Tab. 1, we summarize the inefficiency reductions possible through ConfTr (trained with THRLP) in comparison to Bel (trained with THRL) and the baseline. Bel does *not* consistently improve inefficiency on all datasets. Specifically, on MNIST, EMNIST or CIFAR100, inefficiency actually *worsens*. Our ConfTr, in contrast, reduces inefficiency consistently, not only for THR but also for APS. Here, improvements on CIFAR for THR are generally less pronounced. This is likely because we train linear models on top of a pre-trained ResNet (He et al., 2016) where features are not taking into account conformalization at test time, see App. J. For APS, in contrast, improvements are still significant. Across all datasets, training with $\mathcal{L}_{\text{class}}$ generally performs slightly better, especially for datasets with many classes such as EMNIST ($K{=}52$) or CIFAR100 ($K{=}100$). Overall, ConfTr yields significant inefficiency reductions, independent of the CP method used at test time.

**Conformal Predictors for Training:** In Tab. 1, we use THRLP during training, irrespective of the CP method used at test time. This is counter-intuitive when using, *e.g.*, APS at test time. However, training with THR and APS is rather difficult, as discussed in App. I. This is likely caused by limited gradient flow as both THR and APS are defined on the predicted probabilities instead of log-probabilities as used for THRLP or in cross-entropy training. Moreover, re-formulating the conformity scores of APS in Eq. (11) to use log-probabilities is non-trivial. In contrast, Bel has to be trained using THRL as a fixed threshold $\tau$ is used during training. This is because the calibration step is ignored during training. Also, fixing $\tau$ is not straightforward for THR due to the limited range of the predicted probabilities $\pi_{\theta,k}(x) \in [0,1]$, see App. E. We believe that this contributes to the poor performance of Bel on several datasets. Finally, we found that Bel or ConfTr do not necessarily recover the accuracy of the baseline. Remember that we refer to the accuracy in terms of the $\arg\max$-prediction of $\pi_\theta$. When training from scratch, accuracy can be 2-6% lower while still *reducing* inefficiency. This is interesting because ConfTr is still able to improve inefficiency, highlighting that cross-entropy training is not appropriate for CP.

**Further Results:** Tab. 2 includes additional results for ConfTr to "conformalize" ensembles on CIFAR10 (left) and with lower confidence levels $\alpha$ on EMNIST (right). In the first example, we consider applying CP to an ensemble of models. Ensemble CP methods such as (Yang & Kuchibhotla, 2021) cannot improve Ineff over the best model of the ensemble, *i.e.*, 3.10 for THR. Instead,

Table 2: **Ensemble Results and Lower Confidence Levels** $\alpha$: *Left*: "Conformalization" of ensembles using a 2-layer MLP trained on logits, either normally or using ConfTr. The ensemble contains 18 models with accuracies in between $75.10$ and $82.72\%$. Training a model on top of the ensemble clearly outperforms the best model of the ensemble; using ConfTr further boosts Ineff. *Right*: The inefficiency improvements of Tab. 1 generalize to lower confidence levels $\alpha$ on EMNIST, although ConfTr is trained with $\alpha{=}0.01$.

| **CIFAR10:** Ensemble Results | | | | **EMNIST:** Confidence Levels | | |
|---|---|---|---|---|---|---|
| Test | THR | | | Method | Basel. | ConfTr |
| Method | (Models) | +MLP | +ConfTr | Test | | THR |
| Avg. Ineff | 3.10 | 2.40 | **2.35** | Ineff, $\alpha{=}0.005$ | 4.10 | **3.37** (-17.8%) |
| Best Ineff | 2.84 | 2.33 | **2.30** | Ineff, $\alpha{=}0.001$ | 15.73 | **13.65** (-13.2%) |

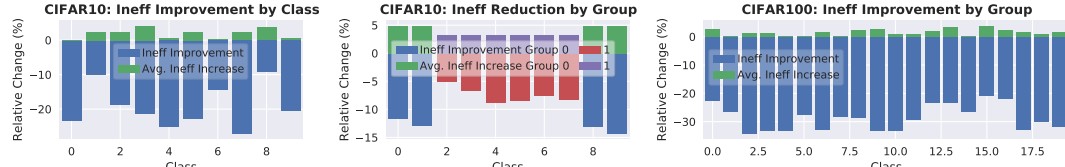

Figure 3: **Shaping Class-Conditional Inefficiency on CIFAR:** Possible inefficiency reductions, in percentage change, per class (blue) and the impact on the overall, average inefficiency across classes (green). *Left:* Significant inefficiency reductions are possible for all classes on CIFAR10. *Middle:* The same strategy applies to groups of classes, *e.g.*, "vehicles" vs "animals", as well. *Right:* Similarly, on CIFAR100, we group classes by their coarse class (20 groups à 5 classes), see (Krizhevsky, 2009), allowing inefficiency improvements of more than 30% per individual group.

training an MLP on top of the ensemble's logits can improve Ineff to 2.40 and additionally using ConfTr to 2.35. The second example shows that ConfTr, trained for $\alpha=0.01$, generalizes very well to significantly smaller confidence levels, *e.g.*, $\alpha=0.001$ on EMNIST. In fact, the improvement of ConfTr (without $\mathcal{L}_{\text{class}}$) in terms of inefficiency is actually more significant for lower confidence levels. We also found ConfTr to be very stable regarding hyper-parameters, see App. H. Only too small batch sizes (*e.g.*, $|B|=100$ on MNIST) prevents convergence. This is likely because of too few examples ($|B_{\text{cal}}|=50$) for calibration with $\alpha=0.01$ during training. More results, *e.g.*, on binary datasets or including additional hyper-parameter ablation can be found in App. J.

## 4.2 CONFORMAL TRAINING FOR APPLICATIONS: CASE STUDIES

For the second part, we focus on ConfTr trained with THRLP and evaluated using THR. We follow Sec. 3.3 and start by reducing class- or group-conditional inefficiency using ConfTr (without $\mathcal{L}_{\text{class}}$), before demonstrating reductions in coverage confusion of two or more classes and avoiding mis-coverage between groups of classes (with $\mathcal{L}_{\text{class}}$). Because this level of control over the confidence sets is not easily possible using Bel or standard CP, we concentrate on ConfTr only:

**Shaping Conditional Inefficiency:** We use ConfTr to reduce class-conditional inefficiency for specific classes or a group of classes, as defined in Eq. (6). In Fig. 2, inefficiency is shown to vary widely across classes: On CIFAR10, the more difficult class 3 ("cat") obtains higher inefficiency than the easier class 1 ("automobile"). Thus, in Fig. 3, we use $\omega=10$ as described in Sec. 3.3 to reduce class- or group-conditional inefficiency. We report the *relative* change in percentage, showing that inefficiency reductions of 20% or more are possible for many classes, including "cat" on CIFAR10 (left, blue). This is also possible for two groups of classes, "vehicles" vs. "animals" (middle). However, these reductions usually come at the cost of a slight increase in average inefficiency across all classes (green). On CIFAR100, we consider 20 coarse classes, each containing 5 of the 100 classes (right). Again, significant inefficiency reductions per coarse class are possible. These observations generalize to all other considered datasets and different class groups, see App. L.

**Avoiding Coverage Confusion:** Next, we use ConfTr to manipulate the coverage confusion matrix as defined in Eq. (7). Specifically, we intend to reduce coverage confusion of selected sets of classes.

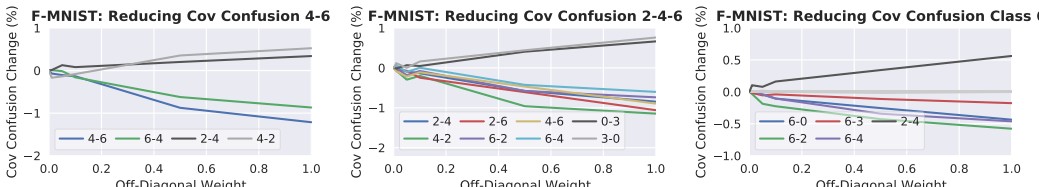

Figure 4: **Controlling Coverage Confusion:** Controlling coverage confusion using ConfTr with $\mathcal{L}_{\text{class}}$ and an increasing penalty $L_{y,k}>0$ on Fashion-MNIST. For classes 4 and 6 ("coat" and "shirt"), coverage confusion $\Sigma_{y,k}$ and $\Sigma_{k,y}$ decreases significantly (blue and green). However, confusion of class 4 with class 2 ("pullover") might increase (gray). ConfTr can also reduce coverage confusion of multiple pairs of classes (*e.g.*, additionally considering class 2). Instead, we can also penalize confusion for each pair $(y,k)$, $k \in [K]$, *e.g.*, $y=6$. Here, $L_{y,k}>0$, but $L_{y,k}=0$, *i.e.*, Cover confusion is not reduced symmetrically.

| CIFAR10: $K_0 = 3$ ("cat") vs. $K_1 =$ Others CIFAR100: $K_0 =$ "human-made vs. $K_1 =$ "natural" | | | | | | |
|---|---|---|---|---|---|---|
| | | CIFAR10 | | | CIFAR100 | |
| | | MisCover $\downarrow$ | | | MisCover $\downarrow$ | |
| Method | Ineff | $0{\to}1$ | $1{\to}0$ | Ineff | $0{\to}1$ | $1{\to}0$ |
| ConfTr | **2.84** | 98.92 | 36.52 | **10.44** | 40.09 | 29.6 |
| $L_{K_0,K_1}{=}1$ | 2.89 | **91.60** | 34.74 | 16.50 | **15.77** | 70.26 |
| $L_{K_1,K_0}{=}1$ | 2.92 | 97.36 | **26.43** | 11.35 | 45.37 | **17.56** |

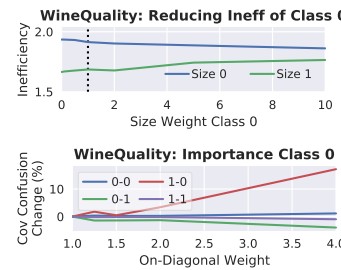

Figure 5: *Left:* **Reducing Mis-Coverage:** Following Sec. 3.3, ConfTr allows to reduce mis-coverage on CIFAR. We consider $K_0 = \{3\}$ (*i.e.*, "cat") vs. all other classes on CIFAR10 (left) and "human-made" vs. "natural" on CIFAR100 ($|K_0|{=}35$, $|K_1|{=}65$, right). On CIFAR10, both MisCover$_{0\to1}$ and MisCover$_{1\to1}$ can be reduced significantly without large impact on inefficiency. For CIFAR100, in contrast, Ineff increases more significantly. *Right:* **Binary Class-Conditional Inefficiency and Coverage:** We plot inefficiency by class (top) and coverage confusion (bottom) on WineQuality. We can reduce inefficiency for class 0 ("bad"), the minority class, at the expense of higher inefficiency for class 1 ("good") and boost class-conditional coverage for class 0.

Using a non-zero entry $L_{y,k} > 0$, $y \neq k$ in $\mathcal{L}_{\text{class}}$, as described in Sec. 3.3, Fig. 4 (left) shows that coverage confusion can be reduced significantly for large enough $L_{y,k}$ on Fashion-MNIST: Considering classes 4 and 6 ("coat" and "shirt") confusion can be reduced by roughly $1\%$. However, as accuracy stays roughly the same and coverage is guaranteed, this comes at the cost of increasing coverage confusion for other class pairs, *e.g.*, 2 ("pullover") and 4. ConfTr can also be used to reduce coverage confusion of multiple class pairs (middle) or a whole row in the coverage confusion matrix $\Sigma_{y,k}$ with fixed $y$ and $y{\neq}k{\in}[K]$. Fig. 4 (right) shows the results for class 6: coverage confusion with, *e.g.*, classes 0 ("t-shirt"), 2 or 4 (blue, green and violet) is reduced roughly $0.5\%$ each at the cost of increased confusion of classes 2 and 4 (in gray). These experiments can be reproduced on other datasets, *e.g.*, MNIST or CIFAR10 in App. M.

**Reducing Mis-Coverage:** We can also address unwanted "overlap" of two groups of classes using ConfTr and $\mathcal{L}_{\text{class}}$. In Fig. 5 (left) we explicitly measure mis-coverage as defined in Eq. (8). First, on CIFAR10, we consider a singleton group $K_0{=}\{3\}$ ("cat") and $K_1{=}[K] \setminus \{3\}$: The ConfTr baseline MisCover$_{0\to1}$ tells us that $98.92\%$ of confidence sets with true class 3 also contain other classes. Given an average inefficiency of $2.84$ this is reasonable. Using $L_{3,k} = 1$, $k \neq 3$, this can be reduced to $91.6\%$. Vice-versa, the fraction of confidence sets of class $y{\neq}3$ containing class 3 can be reduced from $36.52\%$ to $26.43\%$. On CIFAR100, this also allows to reduce overlap between "human-made" (35 classes) and "natural" (65 classes) things, *e.g.*, MisCover$_{0\to1}$ reduces from $40.09\%$ to $15.77\%$, at the cost of a slight increase in inefficiency. See App. N for additional results.

**Binary Datasets:** Finally, in Fig. 5 (right), we illustrate that the above conclusions generalize to the binary case: On WineQuality, we can control inefficiency of class 0 ("bad wine", minority class with $\sim37\%$ of examples) at the expense of increased inefficiency for class 1 ("good wine", top). Similarly, we can (empirically) improve class-conditional coverage for class 0 (bottom) or manipulate coverage confusion of both classes, see App. O.

## 5 CONCLUSION

We introduced **conformal training (ConfTr)**, a novel method to train conformal predictors *end-to-end* with the underlying model. This addresses a major limitation of conformal prediction (CP) in practice: The model is fixed, leaving CP little to no control over the predicted confidence sets. In thorough experiments, we demonstrated that ConfTr can improve inefficiency of state-of-the-art CP methods such as THR (Sadinle et al., 2019) or APS (Romano et al., 2020). More importantly, motivated by medical diagnosis, we highlighted the ability of ConfTr to manipulate the predicted confidence sets in various ways. First, ConfTr can "shape" the class-conditional inefficiency distribution, *i.e.*, reduce inefficiency on specific classes at the cost of higher inefficiency for others. Second, ConfTr allows to control the coverage-confusion matrix by, *e.g.*, reducing the probability of including classes other than the ground truth in confidence sets. Finally, this can be extended to explicitly reduce "overlap" between groups of classes in the predicted confidence sets. In all cases, ConfTr does *not* lose the (marginal) coverage guarantee provided by CP.

## ETHICS STATEMENT

Recent deep learning based classifiers, as used in many high-stakes applications, achieve impressive accuracies on held-out test examples. However, this does *not* provide sufficient guarantees for safe deployment. Conformal prediction (CP), instead, predicts *confidence sets* equipped with a guarantee that the true class is included with specific, user-specified probability. These confidence sets also provide intuitive uncertainty estimates. We specifically expect CP to be beneficial in the medical domain, improving trustworthiness among doctors and patients alike by providing performance guarantees and reliable uncertainty estimates. Yet, the current work does *not* contain experiments with personal/sensitive medical data. The presented results are on standard benchmark datasets only.

However, these benefits of CP may not materialize in many applications unless CP can be better integrated into existing classifiers. These are predominantly deep networks, trained end-to-end to, *e.g.*, optimize classification performance. CP, in contrast, is agnostic to the underlying model, being applied as "wrapper" post-training, such that the obtained confidence sets may not be optimal, *e.g.*, in terms of size (*inefficiency*) or composition (*i.e.*, the included classes). Especially in the medical domain, constraints on the confidence sets can be rather complex. Our *conformal training (ConfTr)* integrates CP into the training procedure, allowing to optimize very specific objectives defined on the predicted confidence sets – without losing the guarantees. In medical diagnosis, smaller confidence sets may avoid confusion or anxiety among doctors or patients, ultimately leading to better diagnoses. For example, we can reduce inefficiency (*i.e.*, the ambiguity of predicted conditions) for conditions that are particularly difficult for doctors to diagnose. Alternatively, ConfTr allows to avoid confusion between low- and high-risk conditions within the confidence sets.

Generally, beyond medical diagnosis, we believe ConfTr to have positive impact in settings where additional constraints on confidence sets are relevant *in addition* to the guarantees and uncertainty estimates provided by CP.

## REPRODUCIBILITY STATEMENT

In order to ensure reproducibility, we include a detailed description of our experimental setup in App. F. We discuss all necessary information for conformal training (ConfTr) as well as our baselines. This includes architectures, training procedure and hyper-parameters, as well as pre-processing/data augmentation if applicable. Furthermore, we describe our evaluation procedure which includes multiple calibration/test splits for conformal prediction (CP) at test time as well as multiple training runs to capture randomness in the used calibration examples and during training. To this end, Tab. A reports the training/calibration/test splits of all used datasets and Tab. B the used hyper-parameters for ConfTr. While Alg. 1 already summarizes the used (smooth) threshold CP methods and our ConfTr, App. P (specifically Alg. B) lists the corresponding Python implementation of these key components.

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

## A  OVERVIEW AND OUTLINE

In the appendix, we discuss an additional baseline, called *coverage training (CoverTr)*, provide additional details on our experimental setup and include complementary results. Specifically, the appendix includes:

- Additional discussion of related work in Sec. B;
- Formal statement of the coverage guarantee obtained through conformal prediction (CP) in Sec. C;
- Differentiable implementation of APS in Sec. D;
- Discussion of *coverage training (CoverTr)* and (Bellotti, 2021) in Sec. E;
- Details on our experimental setup, including dataset splits, model details and used hyper-parameters for ConfTr, in Sec. F;
- Experiments regarding random training and test trials in Sec. G;
- Hyper-parameter ablation on MNIST in Sec. H;
- CoverTr and ConfTr ablation on MNIST and Fashion-MNIST in Sec. I;
- Complete inefficiency (Ineff) results on all datasets in Sec. J;
- Effect of (standard) ConfTr on class-conditional inefficiency and coverage (Cover) confusion in Sec. K;
- Additional results for ConfTr shaping the class-conditional inefficiency distribution in Sec. L;
- More experiments for ConfTr manipulating coverage confusion in Sec. M;
- Complementary results for ConfTr reducing mis-coverage (MisCover) in Sec. N;
- Class-conditional inefficiency and coverage confusion on binary datasets in Sec. O;
- Python- and Jax (Bradbury et al., 2018) code for ConfTr in Sec. P.

## B  RELATED WORK

Conformal prediction (CP) builds on early work by Vovk et al. (2005) considering both regression, see *e.g.*, (Romano et al., 2019) for references, and classification settings, *e.g.* (Romano et al., 2020; Angelopoulos et al., 2021; Cauchois et al., 2020; Hechtlinger et al., 2018). Most of these approaches follow a *split* CP approach (Lei et al., 2013) where a held-out calibration set is used, as in the main paper, however, other variants based on cross-validation (Vovk, 2013) or jackknife (*i.e.*, leave-one-out) (Barber et al., 2019a) are available. These approaches mostly provide marginal coverage. Vovk (2012); Barber et al. (2019b) suggest that it is generally difficult or impossible to obtain conditional coverage. However, Romano et al. (2020) work towards *empirically* better conditional coverage and Sadinle et al. (2019) show that efficient *class-conditional* coverage is possible. Angelopoulos et al. (2021) extend the work by Romano et al. (2020) to obtain smaller confidence sets at the expense of the obtained empirical conditional coverage. CP has also been studied in the context of ensembles (Yang & Kuchibhotla, 2021), allowing to perform model selection based on inefficiency while keeping coverage guarantees. The work of Bates et al. (2021) can be seen as a CP extension in which a guarantee on an arbitrary, user-specified risk can be obtained, using a conformal predictor similar to (Sadinle et al., 2019). Our conformal training (ConfTr) follows the split CP approach and is specifically targeted towards classification problems. Nevertheless, extensions to regression, or other CP formulations such as (Bates et al., 2021) during training, are possible. Beyond that, ConfTr is agnostic to the CP method used at test time and can thus be seen as complementary to the CP methods discussed above. This means that ConfTr can easily be combined with approaches such as (Bates et al., 2021) or class-conditional conformal predictors (Sadinle et al., 2019) at test time.

In terms of *learning* to predict confidence sets, our approach has similarities to the *multiple choice learning* of Guzmán-Rivera et al. (2012) which yields multiple possible outputs in structured prediction settings (*e.g.*, image segmentation). However, the obtained prediction sets are fixed size and no coverage guarantee is provided. Concurrent work by Bellotti (2021) is discussed in detail in App. E.

## C  COVERAGE GUARANTEE

Following Romano et al. (2020), we briefly state the coverage guarantee obtained by CP in formal terms: Given that the learning algorithm used is invariant to permutations of the training examples, and the calibration examples $\{(X_i, Y_i)\}_{i \in I_{\text{cal}}}$ are exchangeably drawn from the same distribution encountered at test time, the discussed CP methods satisfy

$$P(Y \in C(X)) \geq 1 - \alpha. \tag{9}$$

As highlighted in (Romano et al., 2020), this bound is near tight if the scores $E(x_i)$ are almost surely distinct:

$$P(Y \in C(X)) \leq 1 - \alpha + \frac{1}{|I_{\text{cal}}| + 1}. \tag{10}$$

Note that this is the case for APS due to the uniform random variable $U$ in Eq. (11). (Romano et al., 2020) notes that there is generally no guarantee on conditional coverage, as this requires additional assumptions. However, class-conditional coverage can be obtained using THR as outlined in (Sadinle et al., 2019). Moreover, Sadinle et al. (2019) show that THR is the most efficient conformal predictor given a fixed model $\pi_\theta$, *i.e.*, minimizes inefficiency. We refer to (Sadinle et al., 2019) for exact statements of the latter two findings.

## D  DIFFERENTIABLE APS

Our differentiable implementation closely follows the one for THR outlined in Sec. 2.2. The main difference is the conformity score $E(x, k)$ computation, *i.e.*,

$$E_\theta(x, k) := \pi_{\theta, y^{(1)}}(x) + \ldots + \pi_{\theta, y^{(k-1)}}(x) + U\pi_{\theta, y^{(k)}}(x), \tag{11}$$

where $\pi_{\theta, y^{(1)}}(x) \geq \ldots \geq \pi_{\theta, y^{(K)}}(x)$ and $U$ is a uniform random variable in $[0, 1]$ to break ties. As in the calibration step, we use an arbitrary smooth sorting approach for this. This implementation could easily be extended to include the regularizer of Angelopoulos et al. (2021), as well.

## E  COVERAGE TRAINING

As intermediate step towards *conformal training (ConfTr)*, we can also ignore the calibration step and just differentiate through the prediction step, *i.e.*, $C_\theta(X; \tau)$. This can be accomplished by fixing the threshold $\tau$. Then, $\pi_\theta$ essentially learns to produce probabilities that yield "good" confidence sets $C_\theta(X; \tau)$ for the chosen threshold $\tau$. Following Alg. A, *coverage training (CoverTr)* computes $C_\theta(X; \tau)$ on each mini-batch using a fixed $\tau$. The model's parameters $\theta$ are obtained by solving

$$\min_\theta \log \left( \mathbb{E} \left[ \mathcal{L}(C_\theta(X; \tau), Y) + \lambda \Omega(C_\theta(X; \tau)) \right] \right). \tag{12}$$

Again, $\mathcal{L}$ is the classification loss from Eq. (5) and $\Omega$ the size loss from Eq. (3). The classification loss has to ensure that the true label $y$ is in the predicted confidence set $C_\theta(X; \tau)$ as the calibration step is missing. In contrast to ConfTr, CoverTr strictly requires both classification and size loss during training. This is because using a fixed threshold $\tau$ yields trivial solutions for both classification and size loss when used in isolation (*i.e.*, $\mathcal{L}$ is minimized for $C_\theta(X; \tau) = [K]$ and $\Omega$ is minimized for $C_\theta(X; \tau) = \emptyset$). Thus, balancing both terms in Eq. (12) using $\lambda$ is crucial during training. As with ConfTr, the threshold $\tau$ is re-calibrated at test time to obtain a coverage guarantee. Choosing $\tau$ for training, in contrast, can be difficult: First, $\tau$ will likely evolve during training (when $\pi_\theta$ gets

| |
|---|
| 1: **function** COVERAGETRAINING$(\tau, \lambda)$ |
| 2:   **for** mini-batch $B$ **do** |
| 3:     $C(x_i; \tau) := \text{SMOOTHPRED}(\pi_\theta(x_i), \tau), i \in B$ |
| 4:     $\mathcal{L}_B := \sum_{i \in B} \mathcal{L}(C_\theta(x_i; \tau), y_i)$ |
| 5:     $\Omega_B := \sum_{i \in B} \Omega(C_\theta(x_i; \tau))$ |
| 6:     $\Delta := \nabla_\theta 1/|B|(\mathcal{L}_B + \lambda \Omega_B)$ |
| 7:     update parameters $\theta$ using $\Delta$ |

Algorithm A: **Coverage Training (CoverTr)**: Compared to Alg. 1 for ConfTr, CoverTr simplifies training by not differentiating through the calibration step and avoiding splitting the batch $B$ in half. However, fixing the threshold $\tau$ can be a problem and training requires both coverage and size loss.

Table A: **Used Datasets:** Summary of train/calibration/test splits, epochs and models used on all datasets in our experiments. The calibration set is usually less than 10% of the training set. On most datasets, the test set is roughly two times larger than the calibration set. When computing random calibration/test splits for evaluation, see text, the number of calibration and test examples stays constant. * On Camelyon, we use features provided by Wilder et al. (2020) instead of the original images. ** For EMNIST, we use a custom subset of the "byClass" split.

| Dataset Statistics | | | | | | | |
|---|---|---|---|---|---|---|---|
| Dataset | Train | Cal | Test | Dimensions | Classes | Epochs | Model |
| Camelyon2016* (Bejnordi et al., 2017) | 280 | 100 | 17 | 31 | 2 | 100 | 1-layer MLP |
| GermanCredit (Dua & Graff, 2017) | 700 | 100 | 200 | 24 | 2 | 100 | Linear |
| WineQuality (Cortez et al., 2009) | 4500 | 500 | 898 | 11 | 2 | 100 | 2-layer MLP |
| MNIST (LeCun et al., 1998) | 55k | 5k | 10k | $28 \times 28$ | 10 | 50 | Linear |
| EMNIST** (Cohen et al., 2017) | 98.8k | 5.2k | 18.8k | $28 \times 28$ | 52 | 75 | 2-layer MLP |
| Fashion-MNIST (Xiao et al., 2017) | 55k | 5k | 10k | $28 \times 28$ | 10 | 150 | 2-layer MLP |
| CIFAR10 (Krizhevsky, 2009) | 45k | 5k | 10k | $32 \times 32 \times 3$ | 10 | 150 | ResNet-34 |
| CIFAR100 (Krizhevsky, 2009) | 45k | 5k | 10k | $32 \times 32 \times 3$ | 100 | 150 | ResNet-50 |

more and more accurate) and, second, the general ballpark of reasonable thresholds $\tau$ depends on the dataset as well as model and is difficult to predict in advance.

In concurrent work by Bellotti (2021) (referred to as Bel), the problem with fixing a threshold $\tau$ is circumvented by using THRL during training, *i.e.*, THR on logits. As the logits are unbounded, the threshold can be chosen arbitrarily, *e.g.*, $\tau = 1$. As Bel also follows the formulation of Eq. (12), the approach can be seen as a special case of CoverTr. However, a less flexible *coverage loss* is used during training: Instead of $\mathcal{L}_{\text{class}}$, the loss is meant to enforce a specific coverage level $(1 - \alpha)$ on each mini-batch. This is done using a squared loss on coverage:

$$\mathcal{L}_{\text{cov}} := \left[ \left( \frac{1}{|B|} \sum_{i \in B} C_{\theta, y_i}(x_i; \tau) \right) - (1 - \alpha) \right]^2 \tag{13}$$

for a mini-batch $B$ of examples. In contrast to Eq. (12), $\mathcal{L}_{\text{cov}}$ is applied per batch and not per example. For the size loss, Bellotti (2021) uses $\kappa = 0$ in Eq. (3). Besides *not* providing much control over the confidence sets, $\mathcal{L}_{\text{cov}}$ also encourages coverage $(1 - \alpha)$ instead of perfect coverage. Nevertheless, this approach is shown to improve inefficiency of THRL on various UCI datasets (Dua & Graff, 2017) using linear logistic regression models. The experiments in the main paper show that this generalizes to non-linear models and more complex datasets. Nevertheless, Bel is restricted to THRL which is outperformed significantly by both THR and APS. Thus, Bel is consistently outperformed by ConfTr in terms of inefficiency improvements. Moreover, the approach *cannot* be used for any of the studied use cases in Sec. 3.3.

Using CoverTr with THR and APS remains problematic. While we found $\tau \in [0.9, 0.99]$ (or $[-0.1, -0.01$ for THRLP) to work reasonably on some datasets, we had difficulties on others, as highlighted in Sec. I. Moreover, as CoverTr requires balancing coverage $\mathcal{L}$ and size loss $\Omega$, hyper-parameter optimization is more complex compared to ConfTr. By extension, these problems also limit the applicability of Bel. Thus, we would ideally want to re-calibrate the threshold $\tau$ after each model update. Doing calibration on a larger, held-out calibration set, however, wastes valuable training examples and compute resources. Thus, ConfTr directly calibrates on each mini-batch and also differentiates through the calibration step itself to obtain meaningful gradients.

# F  EXPERIMENTAL SETUP

**Datasets and Splits:** We consider Camelyon2016 (Bejnordi et al., 2017), GermanCredit (Dua & Graff, 2017), WineQuality (Cortez et al., 2009), MNIST (LeCun et al., 1998), EMNIST (Cohen et al., 2017), Fashion-MNIST (Cohen et al., 2017) and CIFAR (Krizhevsky, 2009) with a fixed split of training, calibration and test examples. Tab. A summarizes key statistics of the used datasets which we elaborate on in the following. Except Camelyon, all datasets are provided by Tensorflow (Abadi et al., 2015)[1]. For Camelyon, we use the pre-computed features of Wilder et al. (2020) which

---

[1]`https://www.tensorflow.org/datasets`

are based on open source code from the Camelyon2016 challenge[2]. For datasets providing a default training/test split, we take the last 10% of training examples as calibration set. On Camelyon, we use the original training set, but split test examples into 100 validation and 17 test examples. This is because less than 100 calibration examples are not meaningful for $\alpha{=}0.05$. As we evaluate 10 random calibration/test splits, the few test examples are not problematic in practice. On GermanCredit and WineQuality, we manually created training/calibration/test splits, roughly matching 70%/10%/20%. We use the "white wine" subset for WineQuality; to create a binary classification problem, wine with quality 6 or higher is categorized as "good wine" (class 1), following (Bellotti, 2021). Finally, for EMNIST, we consider a subset of the "byClass" split that contains $52 = 2 \cdot 26$ classes comprised of all lower and upper case letters. We take the first 122.8k examples, split as in Tab. A.

**Models and Training:** We consider linear models, multi-layer perceptrons (MLPs) and ResNets (He et al., 2016) as shown in Tab. A. Specifically, we use a linear model on MNIST and German-Credit, 1- or 2-layer MLPs on Camelyon2016, WineQuality and Fashion-MNIST, and ResNet-34/50 (He et al., 2016) on CIFAR10/100. Models and training are implemented in Jax (Bradbury et al., 2018)[3] and the ResNets follow the implementation and architecture provided by Haiku (Hennigan et al., 2020)[4]. Our $l$-layer MLPs comprise $l$ *hidden* layers. We use 32, 256, 128, 64 units per hidden layer on Camelyon, WineQuality, EMNIST and Fashion-MNIST, respectively. These were chosen by grid search over $\{16, 32, 64, 128, 256\}$. In all cases, we use ReLU activations (Nair & Hinton) and batch normalization (Ioffe & Szegedy, 2015). We train using stochastic gradient descent (SGD) with momentum 0.0005 and Nesterov gradients. The baseline models are trained with cross-entropy loss, while ConfTr follows Alg. 1 and CoverTr follows Alg. A. Learning rate and batch size are optimized alongside the ConfTr hyper-parameters using grid search, see below. The number of epochs are listed in Tab. A and we follow a multi-step learning rate schedule, multiplying the initial learning rate by 0.1 after $^2/_5$, $^3/_5$ and $^4/_5$ of the epochs. We use Haiku's default initializer. On CIFAR, we apply whitening using the per-channel mean and standard deviation computed on the training set. On the non-image datasets (Camelyon, GermanCredit, WineQuality), we whiten each feature individually. On MNIST, EMNIST and Fashion-MNIST, the input pixels are just scaled to $[-1, 1]$. Except on CIFAR, see next paragraph, we do *not* use any data augmentation. Finally, we do *not* use Platt scaling (Guo et al., 2017) as used in (Angelopoulos et al., 2021).

**Fine-Tuning on CIFAR:** On CIFAR10 and CIFAR100, we train base ResNet-34/ResNet-50 models which are then fine-tuned using Bel, CoverTr or ConfTr. We specifically use a ResNet-34 with only 4 base channels to obtain an accuracy of 82.6%, using only random flips and crops as data augmentation. The rationale is to focus on the results for CP at test time, without optimizing accuracy of the base model. On CIFAR100, we use 64 base channels for the ResNet-50 and additionally employ AutoAugment (Cubuk et al., 2018) and Cutout (Devries & Taylor, 2017) as data augmentation. This model obtains 73.64% accuracy. These base models are trained on 100% of the training examples (without calibration examples). For fine-tuning, the last layer (*i.e.*, logit layer) is re-initialized and trained using the same data augmentation as applied for the base model, subject to the random training trials described below. We also consider "extending" the ResNet by training a 2-layer MLP with 128 units per hidden layer on top of the features (instead of re-initializing and fine-tuning the logit layer). All reported results either correspond to fine-tuned (*i.e.*, linear model on features) or extended models (*i.e.*, 2-layer MLP on features) trained on these base models.

**Hyper-Parameters:** The final hyper-parameters selected for ConfTr (for THR at test time) on all datasets are summarized in Tab. B. These were obtained using grid search over the following hyper-parameters: batch size in $\{1000, 500, 100\}$ for WineQuality, MNIST, EMNIST, Fashion-MNIST and CIFAR, $\{300, 200, 100, 50\}$ on GermanCredit and $\{80, 40, 20, 10\}$ on Camelyon; learning rate in $\{0.05, 0.01, 0.005\}$; temperature $T \in \{0.01, 0.1, 0.5, 1\}$; size weight $\lambda \in \{0.0001, 0.0005, 0.001, 0.005, 0.01, 0.05, 0.1, 0.5, 1, 5, 10\}$ (*c.f.* Eq. (1), right); and $\kappa \in \{0, 1\}$ (*c.f.* Eq. (3)). Grid search was done for each dataset individually on 100% of the training examples (*c.f.* Tab. A). That is, for hyper-parameter optimization we did *not* perform random training trials as described next. The best hyper-parameters according to inefficiency after evaluating 3 random calibration/test splits were selected, both for THR and APS at test time, with and without $\mathcal{L}_{\text{class}}$.

---

[2]`https://github.com/arjunvekariyagithub/camelyon16-grand-challenge`
[3]`https://github.com/google/jax`
[4]`https://github.com/deepmind/dm-haiku`

Table B: **Used ConfTr Hyper-Parameters** with and without $\mathcal{L}_{\text{class}}$ for THRLP during training and THR at test time. The hyper-parameters for APS at test time might vary slightly from those reported here. The exact grid search performed to obtained these hyper-parameters can be found in the text. Note that, while hyper-parameters fluctuate slightly, $\lambda$ needs to be chosen higher when training with $\mathcal{L}_{\text{class}}$. Additionally, and in contrast to Bel, $\kappa = 1$ in Eq. (3) performs better, especially combined with $\mathcal{L}_{\text{class}}$. Note that dispersion for smooth sorting is fixed to $\epsilon = 0.1$.

| **ConfTr Hyper-Parameters** (for THRLP during training and THR at test time) | | | | | |
|---|---|---|---|---|---|
| Dataset, Method | Batch Size | Learning rate | Temp. $T$ | Size weight $\lambda$ | $\kappa$ in Eq. (3) |
| Camelyon, ConfTr | 20 | 0.005 | 0.1 | 5 | 1 |
| Camelyon, ConfTr $+\mathcal{L}_{\text{class}}$ | 10 | 0.01 | 0.01 | 5 | 1 |
| GermanCredit, ConfTr | 200 | 0.05 | 1 | 5 | 1 |
| GermanCredit, ConfTr $+\mathcal{L}_{\text{class}}$ | 400 | 0.05 | 0.1 | 5 | 1 |
| WineQuality, ConfTr | 100 | 0.005 | 0.5 | 0.05 | 1 |
| WineQuality, ConfTr $+\mathcal{L}_{\text{class}}$ | 100 | 0.005 | 0.1 | 0.5 | 1 |
| MNIST, ConfTr | 500 | 0.05 | 0.5 | 0.01 | 1 |
| MNIST, ConfTr $+\mathcal{L}_{\text{class}}$ | 100 | 0.01 | 1 | 0.5 | 1 |
| EMNIST, ConfTr | 100 | 0.01 | 1 | 0.01 | 1 |
| EMNIST, ConfTr $+\mathcal{L}_{\text{class}}$ | 100 | 0.01 | 1 | 5 | 1 |
| Fashion-MNIST, ConfTr | 100 | 0.01 | 0.1 | 0.01 | 0 |
| Fashion-MNIST, ConfTr $+\mathcal{L}_{\text{class}}$ | 100 | 0.01 | 0.1 | 0.5 | 1 |
| CIFAR10, fine-tune ConfTr | 500 | 0.01 | 1 | 0.05 | 0 |
| CIFAR10, fine-tune ConfTr $+\mathcal{L}_{\text{class}}$ | 500 | 0.05 | 0.1 | 1 | 1 |
| CIFAR10, "extend" ConfTr | 100 | 0.01 | 1 | 0.005 | 0 |
| CIFAR10, "extend" ConfTr $+\mathcal{L}_{\text{class}}$ | 500 | 0.05 | 0.1 | 0.1 | 1 |
| CIFAR100, fine-tune ConfTr | 100 | 0.005 | 1 | 0.005 | 0 |
| CIFAR100, fine-tune ConfTr $+\mathcal{L}_{\text{class}}$ | 100 | 0.005 | 1 | 0.01 | 1 |

Table C: **Importance of Random Trials:** We report coverage and inefficiency with the corresponding standard deviation across 10 *test* (left) and 10 *training* trials (right). ConfTr was trained using THRLP if not stated otherwise. For test trials, a fixed model is used. Results for training trials additionally include 10 test trials, but the standard deviation is reported only across the training trials. These results help to disentangle the impact of test and training trials. For example, while ConfTr with APS (during training) works in the best case, the standard deviation of 3.1 across multiple training trials indicates that training is *not* stable.

| **MNIST:** *test* trials, Cover/Ineff for THR | | | | **MNIST:** *Training* trials, Cover/Ineff for THR | | | |
|---|---|---|---|---|---|---|---|
| Method | Acc | Cover | Ineff | Method | Acc | Cover | Ineff |
| Baseline | 92.45 | 99.09±0.2 | 2.23±0.15 | Baseline | 92.4±0.06 | 99.09±0.8 | 2.23±0.01 |
| ConfTr | 90.38 | 99.05±0.2 | 2.14±0.13 | ConfTr | 90.2±0.12 | 99.03±0.22 | 2.18±0.025 |
| ConfTr $+\mathcal{L}_{\text{class}}$ | 91.14 | 99.03±0.19 | 2.09±0.12 | ConfTr $+\mathcal{L}_{\text{class}}$ | 91.2±0.05 | 99.05±0.21 | 2.11±0.028 |
| | | | | ConfTr with APS | 87.9±4.81 | 99.09±0.29 | 5.79±3.1 |

Tab. B allows to make several observations. First, on the comparably small (and binary) datasets Camelyon and GermanCredit, the size weight $\lambda = 5$ is rather high. For ConfTr without $\mathcal{L}_{\text{class}}$, this just indicates that a higher learning rate could be used. Then using $\mathcal{L}_{\text{class}}$, however, this shows that the size loss is rather important for ConfTr, especially on binary datasets. Second, we found the temperature $T$ to have low impact on results, also see Sec. H. On multiclass datasets, the size weight $\lambda$ is usually higher when employing $\mathcal{L}_{\text{class}}$. Finally, especially with $\mathcal{L}_{\text{class}}$, using "valid" size loss, *i.e.*, $\kappa = 1$, to not penalize confidence sets of size 1, works better than $\kappa = 0$.

**Random Training and Test Trials:** For statistically meaningful results, we perform random *test* and *training* trials. Following common practice (Angelopoulos et al., 2021), we evaluate CP methods at test time using 10 random calibration/test splits. To this end, we throw all calibration and test examples together and sample a new calibration/test split for each trial, preserving the original calibration/test composition which is summarized in Tab. A. Metrics such as coverage and inefficiency are then empirically evaluated as the average across all test trials. Additionally, and in contrast to (Bellotti, 2021), we consider random training trials: After hyper-parameters optimization on all training examples, we train 10 models with the final hyper-parameters on a new training set obtained by sampling the original one with up to 5 replacements. For example, on MNIST, with 55k training

Table D: **Hyper-Parameter Ablation on MNIST:** For ConfTr without $\mathcal{L}_{\text{class}}$, we report inefficiency and accuracy when varying hyper-parameters individually: batch size/learning rate, size weight $\lambda$, temperature $T$ and confidence level $\alpha$. While size weight $\lambda$ and temperature $T$ have insignificant impact, too small batch size can prevent ConfTr from converging. Furthermore, the chosen hyper-parameters do not generalize well to higher confidence levels $\alpha \in \{0.1, 0.05\}$.

| Batch Size and Learning Rate | | | | | | | | |
|---|---|---|---|---|---|---|---|---|
| Batch Size | 1000 | 1000 | 1000 | **500** | 500 | 500 | 100 | 100 | 100 |
| Learning Rate | 0.05 | 0.01 | 0.005 | **0.05** | 0.01 | 0.005 | 0.05 | 0.01 | 0.005 |
| Ineff | 2.27 | 2.24 | 2.24 | 2.18 | 2.18 | **2.17** | 8.04 | 7.32 | 9.66 |
| Acc | 89.05 | 89.18 | 89.06 | 90.23 | 90.22 | **90.27** | 11.5 | 22.46 | 12.13 |

| Size Weight $\lambda$ | | | | | | | |
|---|---|---|---|---|---|---|---|
| $\lambda$ | 0.001 | 0.005 | **0.01** | 0.05 | 0.1 | 1 | 10 |
| Ineff | 2.18 | 2.18 | 2.18 | 2.19 | 2.19 | 2.19 | **2.16** |
| Acc | 90.2 | 20.23 | 90.23 | 90.2 | 90.25 | 90.23 | **90.26** |

| Temperature $T$ | | | | | | | |
|---|---|---|---|---|---|---|---|
| $T$ | 0.01 | 0.05 | 0.1 | **0.5** | 1 | 5 | 10 |
| Ineff | 2.39 | 2.23 | 2.2 | 2.19 | **2.18** | 2.2 | 2.29 |
| Acc | 88.54 | 89.94 | 90.02 | 90.24 | **90.28** | 90.05 | 89.63 |

| Confidence Level $\alpha$ (during training) | | | |
|---|---|---|---|
| $\alpha$ | 0.1 | 0.05 | **0.01** | 0.005 |
| Ineff | 8.07 | 7.23 | 2.18 | **2.17** |
| Acc | 12.88 | 39.82 | **90.23** | 89.47 |

examples, we randomly sample 10 training sets of same size with each, on average, containing only ∼68% unique examples from the original training set. Overall, this means that we report, *e.g.*, inefficiency as average over a total of $10 \cdot 10 = 100$ random training *and* test trials. As a consequence, our evaluation protocol accounts for randomness at test time (*i.e.*, regarding the calibration set) and at training time (*i.e.*, regarding the training set, model initialization, *etc.*).

## G  IMPORTANCE OF RANDOM TRIALS

In Tab. C we highlight the importance of random training and test trials for evaluation. On the left, we show the impact of trials at test time, *i.e.*, 10 random calibration/test splits, for a fixed model on MNIST. While the standard deviation of coverage is comparably small, usually $\leq 0.2\%$, standard deviation of inefficiency is higher in relative terms. This makes sense as coverage is guaranteed, while inefficiency depends more strongly on the sampled calibration set. The right table, in contrast, shows that training trials exhibit lower standard deviation in terms of inefficiency. However, training with, *e.g.*, APS will mainly result in high inefficiency, on average, because of large standard deviation. In fact, ConfTr with APS or THR at training time results in worse inefficiency mainly because training is less stable. This supports the importance of running multiple training trials for ConfTr.

## H  IMPACT OF HYPER-PARAMETERS

In Tab. D, we conduct ablation for individual hyper-parameters of ConfTr with THRLP and without $\mathcal{L}_{\text{class}}$ on MNIST. The hyper-parameters used in the main paper, *c.f.* Tab. B, are highlighted in **bold**. As outlined in Sec. F, hyper-parameter optimization was conducted on 100% training examples with only 3 random test trials, while Tab. D shows results using random training *and* test trials. We found batch size and learning rate to be most impactful. While batch sizes 1000 and 500 both work, batch size 100 prevents ConfTr from converging properly. This might be due to the used $\alpha = 0.01$ which might be too low for batch size 100 where only 50 examples are available for calibration during training. Without $\mathcal{L}_{\text{class}}$, the size weight $\lambda$ merely scales the learning rate and, thus, has little to no impact. For ConfTr *with* $\mathcal{L}_{\text{class}}$, we generally found the size weight $\lambda$ to be more important for balancing classification loss $\mathcal{L}$ and size loss $\Omega$ in Eq. (4). Temperature has no significant impact, although a temperature of 0.5 or 1 works best. Finally, the hyper-parameters do generalize to a lower confidence level $\alpha = 0.005$. Significantly lower values, *e.g.*, $\alpha = 0.001$, are, however,

Table E: **Ablation for CoverTr and ConfTr on MNIST and Fashion-MNIST:** We report inefficiency and accuracy for (Bellotti, 2021) (Bel), CoverTr and ConfTr considering various CP methods for training and testing. Bel outperforms the baseline when using ThrL, but does not do so for Thr on MNIST. CoverTr with Thr or APS during training is challenging, resulting in high inefficiency (mainly due to large variation among training trials, $c.f.$ Tab. C), justifying our choice of ThrLP for ConfTr. Also CoverTr is unable to improve over the Thr baseline. Similar observations hold on Fashion-MNIST where, however, CoverTr with Thr or APS was not possible.

| **MNIST:** Ablation for CoverTr and ConfTr | | | | | | | | | | | | | |
|---|---|---|---|---|---|---|---|---|---|---|---|---|---|
| Method | Baseline | | | Bel | | CoverTr | | | | ConfTr | | | |
| Train | | | | ThrL | | Thr | APS | ThrLP | | ThrLP | ThrLP | $+\mathcal{L}_{\text{class}}$ | |
| Test | ThrL | Thr | APS | ThrL | Thr | Thr | APS | Thr | APS | Thr | APS | Thr | APS |
| Avg. Ineff | 3.57 | 2.23 | 2.5 | 2.73 | 2.7 | 6.34 | 4.86 | 2.5 | 2.76 | 2.18 | 2.16 | 2.11 | 2.14 |
| Avg. Acc | 92.39 | 92.39 | 92.39 | 81.41 | 90.01 | 83.85 | 88.53 | 92.63 | 92.63 | 90.24 | 90.21 | 91.18 | 91.35 |

| **Fashion-MNIST:** Ablation for CoverTr and ConfTr | | | | | | | | | | | |
|---|---|---|---|---|---|---|---|---|---|---|---|
| Method | Baseline | | | Bel | | CoverTr | | ConfTr | | | |
| Train | | | | ThrL | | Thr | ThrLP | ThrLP | ThrLP | $+\mathcal{L}_{\text{class}}$ | |
| Test | ThrL | Thr | APS | ThrL | Thr | Thr | Thr | Thr | APS | Thr | APS |
| Ineff | 2.52 | 2.05 | 2.36 | 1.83 | 1.9 | 4.03 | 2.69 | 1.69 | 1.82 | 1.67 | 1.73 |
| Acc | 89.16 | 89.16 | 89.16 | 84.29 | 84.61 | 89.23 | 87.48 | 88.86 | 87.43 | 89.23 | 88.69 |

not meaningful due to the batch size of $500$. However, significantly higher confidence levels, *e.g.*, $\alpha = 0.1$ or $\alpha = 0.05$, require re-optimizing the other hyper-parameters.

## I CoverTr and ConfTr Ablation on MNIST and Fashion-MNIST

In Tab. E, we present an ablation for CoverTr, see Sec. E, and ConfTr on MNIST, using a linear model, and Fashion-MNIST, using a 2-layer MLP. Bel is generally able to improve inefficiency of ThrL. Using Thr, however, Bel worsens inefficiency on MNIST significantly, while improving slightly over the baseline on Fashion-MNIST. As a result, the improvement of ConfTr over Bel is also less significant on Fashion-MNIST. Using CoverTr with Thr or APS during training works poorly. As described in Tab. C, this is mainly due to a high variation across training runs, *i.e.*, individual models might work well, but training is not stable enough to get consistent improvements. Thus, on MNIST, inefficiency for CoverTr with Thr and APS is very high. Moreover, on Fashion-MNIST, we were unable to train CoverTr with Thr and APS. Using ThrLP, training with CoverTr works and is reasonably stable, but does not improve over the baseline. It does improve over Bel on MNIST though. As described in the main paper, we suspect the fixed threshold $\tau$ to be problematic. Overall, however, only ConfTr is able to outperform the Thr baseline on both datasets. Here, ConfTr with $\mathcal{L}_{\text{class}}$ works slightly better than without.

## J All Inefficiency Results

Tab. F shows complementary results for ConfTr on CIFAR10, EMNIST and CIFAR100. For results on MNIST and Fashion-MNIST, see Tab. D. On CIFAR10, we also include ConfTr using a 2-layer MLP on top of ResNet features – instead of the linear model used in the main paper. In Tab. F, this is referred to as "extending". However, inefficiency increases slightly compared to re-initializing and training just the (linear) logit layer. This shows that the smaller inefficiency improvements on CIFAR shown in the main paper are not due to the linear model used, but rather caused by the features themselves. We suspect that this is because the features are trained to optimize cross-entropy loss, leaving ConfTr less flexibility to optimize inefficiency. In Tab. G, we consider three binary datasets, *i.e.*, WineQuality, GermanCredit and Camelyon. On binary datasets, ThrL, Thr and APS perform very similar. This already suggests that there is little room for inefficiency improvements. Indeed, ConfTr is not able to improve inefficiency significantly. However, this is partly due to our thorough evaluation scheme: On Camelyon (using $\alpha{=}0.05$), we do *not* report averages across all training trials, but the results corresponding to the best model. This is because sub-sampling the training examples is unreasonable given that there are only $280$ of them. Thus, Camelyon shows that ConfTr *can* improve inefficiency. On WineQuality or GermanCredit, however, this is "hidden" in reporting averages across 10 training runs.

Table F: **Inefficiency and Accuracy on Multiclass Datasets:** Complementing Tab. 1 in the main paper, we include results for CoverTr on CIFAR10. Furthermore, we consider training a non-linear 2-layer MLP on the ResNet features on CIFAR10, $c.f.$ Sec. F, alongside the ensemble results from the main paper. We report inefficiency and accuracy in all cases, focusing on ConfTr in comparison to Bel. On EMNIST, we additionally consider $\alpha = 0.005, 0.001$ (for the baseline and ConfTr only). As in the main paper, ConfTr consistently improves inefficiency of THR and APS.

**CIFAR10:** Fine-Tuning and "Extending"

| | Baselines | | | Bel | CoverTr | ConfTr (Fine-tuning) | | | | "Extend" ConfTr | |
|---|---|---|---|---|---|---|---|---|---|---|---|
| Method | | | | | | | | | | | |
| Train | | | | THRL | THRLP | THRLP | THRLP | +$\mathcal{L}_{\text{class}}$ | | THRLP | +$\mathcal{L}_{\text{class}}$ |
| Test | THRL | THR | APS | THR | THR | THR | APS | THR | APS | THR | THR |
| Ineff | 3.92 | 2.93 | 3.3 | 2.93 | 2.84 | 2.88 | 3.05 | 2.84 | 2.93 | 2.89 | 2.96 |
| Acc | 82.6 | 82.6 | 82.6 | 82.18 | 82.36 | 82.32 | 82.34 | 82.4 | 82.4 | 82.3 | 82.23 |

**CIFAR10:** Ensemble Results

| Method | (Ensemble Models) | | | Ensemble+MLP | | | Ensemble +ConfTr |
|---|---|---|---|---|---|---|---|
| Train | | | | | | | THRLP |
| Test | THRL | THR | APS | THRL | THR | APS | THR |
| Avg. Ineff | 4.19 | 3.1 | 3.48 | 3.12 | 2.4 | 2.77 | 2.35 |
| Best Ineff | 3.74 | 2.84 | 3.17 | 3.0 | 2.33 | 2.71 | 2.3 |
| Avg. Acc | 80.65 | 80.65 | 80.65 | 85.88 | 85.88 | 85.88 | 85.88 |
| Best Acc | 82.58 | 82.58 | 82.58 | 86.01 | 86.01 | 86.01 | 86.02 |

**EMNIST**

| Method | Baselines | | | Bel | | ConfTr | | | |
|---|---|---|---|---|---|---|---|---|---|
| Train | | | | THRL | THRL | THRLP | THRLP | +$\mathcal{L}_{\text{class}}$ | |
| Test | THRL | THR | APS | THRL | THR | THR | APS | THR | APS |
| Ineff | 5.07 | 2.66 | 4.23 | 3.95 | 3.48 | 2.66 | 2.86 | 2.49 | 2.87 |
| Ineff, $\alpha$=0.005 | 9.23 | 4.1 | 6.04 | – | – | 3.37 | – | – | – |
| Ineff, $\alpha$=0.001 | 23.89 | 15.73 | 19.33 | – | – | 13.65 | – | – | – |
| Acc | 83.79 | 83.79 | 83.79 | 80.69 | 80.69 | 77.1 | 77.43 | 77.49 | 78.09 |

**CIFAR100**

| Method | Baselines | | | Bel | ConfTr | | | |
|---|---|---|---|---|---|---|---|---|
| Train | | | | THRL | THRLP | THRLP | +$\mathcal{L}_{\text{class}}$ | |
| Test | THRL | THR | APS | THR | THR | APS | THR | APS |
| Ineff | 19.22 | 10.63 | 16.62 | 10.91 | 10.78 | 12.99 | 10.44 | 12.73 |
| Acc | 73.36 | 73.36 | 73.36 | 72.65 | 72.02 | 72.78 | 73.27 | 72.99 |

# K  EFFECT OF CONFTR ON CLASS-CONDITIONAL INEFFICIENCY AND COVERAGE CONFUSION

Fig. G shows that standard ConfTr (without $\mathcal{L}_{\text{class}}$) does not have a significant influence on the class-conditional inefficiency distribution compared to the baseline. Similarly, ConfTr with $\mathcal{L}_{\text{class}}$ and identity loss matrix $L = I_K$ does not influence coverage confusion besides reducing overall inefficiency. Specifically, on MNIST, Fashion-MNIST and CIFAR10, we show the class-conditional inefficiency distribution (left) as well as the coverage confusion matrices (middle and right) for the baseline and ConfTr. On the left, we consider ConfTr without $\mathcal{L}_{\text{class}}$, and on the right with $\mathcal{L}_{\text{class}}$. As can be seen, only an overall reduction of inefficiency is visible, the distribution of Ineff$[y]$, $c.f.$ Eq. (6), across classes $y$ remains roughly the same. For coverage confusion $\Sigma$ from Eq. (7), the same observation can be made, $i.e.$, an overall reduction of inefficiency also reduces confusion, but the spatial pattern remains the same. Thus, in the main paper and the following experiments, we always highlight the improvement over standard ConfTr, without $\mathcal{L}_{\text{class}}$ for reducing class-conditional inefficiency and with $\mathcal{L}_{\text{class}}$ for changing coverage confusion or improving MisCover.

Table G: **Inefficiency and Accuracy on Binary Datasets.** Experimental results on the binary datasets WineQuality, GermanCredit and Camelyon. While we include APS on WineQuality, we focus on THR on GermanCredit and Camelyon due to slightly lower inefficiency. However, THRL, THR and APS perform very similarly on all tested binary datasets. Generally, ConfTr does not improve significantly over the baseline. * On Camelyon, we report the best results without training trials as sub-sampling the 280 training examples is prohibitively expensive.

| WineQuality | | | | | | | | | |
|---|---|---|---|---|---|---|---|---|---|
| Method | Baselines | | | Bel | CoverTr | ConfTr | | | |
| Train | | | | THRL | THRLP | THRLP | THRLP | +$\mathcal{L}_{\text{class}}$ | |
| Test | THRL | THR | APS | THR | THR | THR | APS | THR | APS |
| Ineff, $\alpha$=0.01 | 1.76 | 1.76 | 1.79 | 1.77 | 1.81 | 1.75 | 1.82 | 1.74 | 1.77 |
| Ineff, $\alpha$=0.05 | 1.48 | 1.49 | 1.53 | 1.57 | 1.50 | 1.51 | – | 1.52 | – |
| Acc | 82.82 | 82.82 | 82.82 | 71.3 | 81.5 | 73.8 | 74.24 | 73.91 | 73.91 |

| GermanCredit | | | | | | |
|---|---|---|---|---|---|---|
| Method | Baselines | | | Bel | ConfTr | |
| Train | | | | THRL | THRLP | +$\mathcal{L}_{\text{class}}$ |
| Test | THRL | THR | APS | THR | THR | THR |
| Ineff | 1.89 | 1.86 | 1.90 | 1.85 | 1.88 | 1.77 |
| Acc | 74.4 | 74.4 | 74.4 | 72.35 | 72.81 | 69.5 |

| Camelyon* $\alpha$=0.05 | | | | | | |
|---|---|---|---|---|---|---|
| Method | Baselines | | | Bel | ConfTr | |
| Train | | | | THRL | THRLP | +$\mathcal{L}_{\text{class}}$ |
| Test | THRL | THR | APS | THR | THR | THR |
| Best Ineff | 1.41 | 1.47 | 1.59 | 1.25 | 1.2 | 1.25 |
| Best Acc | 88 | 88 | 88 | 92 | 91.5 | 85 |

## L    SHAPING CLASS-CONDITIONAL INEFFICIENCY ON OTHER DATASETS

Fig. A and B provide complementary results demonstrating the ability of ConfTr to shape the class- or group-conditional inefficiency distribution. First, Fig. A plots inefficiency for individual classes on CIFAR10 and coarse classes on CIFAR100. In both cases, significant inefficiency reductions are possible for high weights $\omega$ in Eq. (3), irrespective or whether the corresponding (coarse) class has above-average inefficiency to begin with. This means that inefficiency reduction is possible for easier and harder classes alike. Second, Fig. B plots the relative inefficiency changes, in percentage, possible per-class or group on MNIST, Fashion-MNIST and CIFAR100. For CIFAR100, we show only the first 10 classes for brevity. In all cases, significant inefficiency reductions are possible, at the expense of a slight increases in average inefficiency across all classes. Here, MNIST is considerably

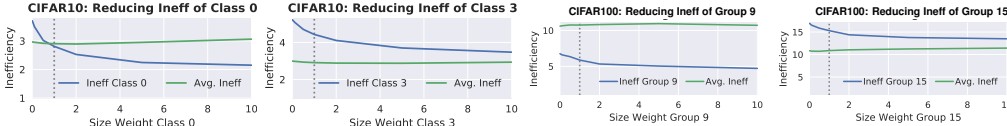

Figure A: **Reducing Class- and Group-Conditional Inefficiency on CIFAR.** Results, complementary to Fig. 3, showing the impact of higher size weights $\omega$ in Eq. (3) for classes 0 and 3 ("airplane" and "cat") on CIFAR10 and coarse classes 9 and 15 ("large man-made outdoor things" and "reptiles") on CIFAR100. ConfTr allows to reduce inefficiency (blue) in all cases, irrespective of whether inefficiency is generally above or below average (green).

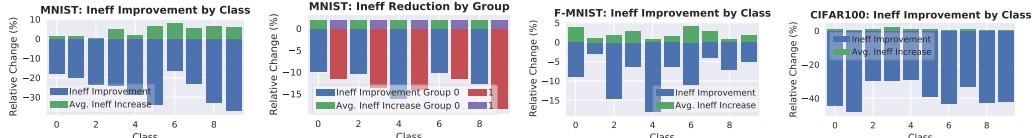

Figure B: **Relative Class and Group-Conditional Inefficiency Improvements:** Complementing the main paper, we plot the possible (relative) inefficiency reduction by class or group ("odd" vs "even") on MNIST and Fashion-MNIST. On CIFAR100, we consider the first 10 classes for brevity. In all cases, significant per-class or -group inefficiency reductions are possible.

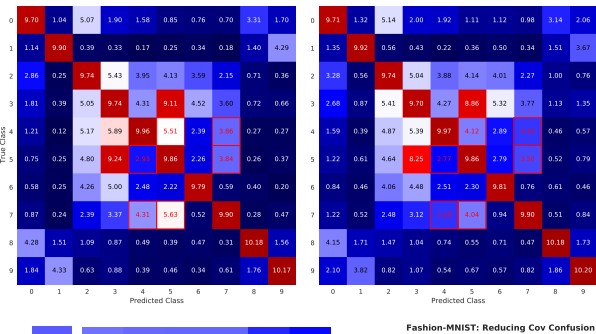

Figure C: **Full Coverage Confusion Matrix on CIFAR10:** We plot the full coverage confusion matrices $\Sigma$ from Eq. (7) on CIFAR10 for the ConfTr baseline (with $\mathcal{L}_{\text{class}}$, left) and ConfTr with $L_{y,k} = 1$ in Eq. (5) for classes $y, k \in \{4, 5, 7\}$ (right, highlighted in red).

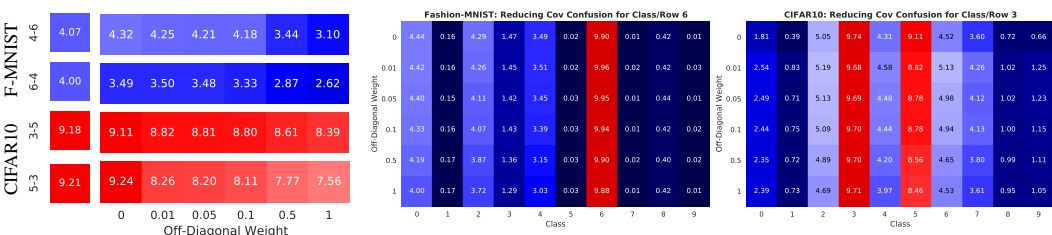

Figure D: **Coverage Confusion Changes on Fashion-MNIST and CIFAR10:** *Left:* coverage confusion change when targeting classes 4 and 6 ("coat" and "shirt") on Fashion-MNIST and 3 and 5 ("cat" and "dog") on CIFAR10. The separate cell on the left is the ConfTr baseline which is, up to slight variations, close to $L_{y,k} = 0$. *Middle and right:* coverage confusion for a whole row, *i.e.*, $\Sigma_{y,k}$ with fixed class $y$ and all $k \neq y$. We show row 6 on Fashion-MNIST and 3 on CIFAR10. In both cases, coverage confusion can be reduced significantly.

easier than Fashion-MNIST: higher inefficiency reductions are possible per class and the cost in terms of average inefficiency increase is smaller. On CIFAR100, inefficiency reductions of 40% or more are possible. This is likely because of the high number of classes, *i.e.*, ConfTr has a lot of flexibility to find suitable trade-offs during training.

# M    Manipulating Coverage Confusion on Other Datasets

Fig. C to E provide additional results for reducing coverage confusion using ConfTr. First, in Fig. C we show the full coverage confusion matrices for the ConfTr baseline (with $\mathcal{L}_{\text{class}}$, left) and ConfTr with $L_{y,k} = 1$, $y \neq k \in \{4, 5, 7\}$ (right, marked in red) on CIFAR10. This allows to get the complete picture of how coverage confusion changes and the involved trade-offs. As demonstrated in the main paper, coverage confusion for, *e.g.*, classes 4 and 5 ("deer" and "dog") reduces. However, coverage confusion for other class pairs might increase slightly. Then, supplementary to Fig. 4 in the main paper, we provide the actual numbers in Fig. D. In particular, we visualize how the actual coverage confusion entries (left) or rows (right) change depending on the off-diagonal weights $L_{y,k}$. Finally, Fig. E presents additional results on MNIST and CIFAR10. From these examples it can be seen that reducing coverage confusion is easier on MNIST, reducing linearly with the corresponding penalty $L_{y,k}$. Moreover, the achieved reductions are more significant. On CIFAR10, in contrast, coverage confusion reduces very quickly for small $L_{y,k}$ before stagnating for larger $L_{y,k}$. At the same time, not all targeted class pairs might yield significant coverage confusion reductions.

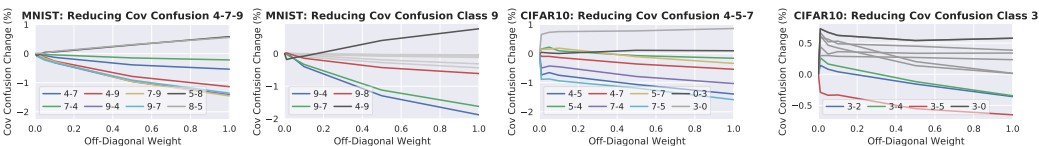

Figure E: **Coverage Confusion Reduction on MNIST and CIFAR10:** Controlling coverage confusion for various class pairs. On MNIST, coverage confusion reduction is usually more significant and the reduction scales roughly linear with the weight $L_{y,k}$. On CIFAR10, in contrast, coverage confusion cannot always be reduced for multiple class pairs at the same time (see light gray).

Table H: **Mis-Coverage on MNIST, Fashion-MNIST and CIFAR10:** We present inefficiency and mis-coverage for various cases: On MNIST, we consider 2 vs. other classes as well as even vs. odd classes. In both cases, mis-coverage can be reduced significantly. As in the main paper, however, reducing $\text{MisCover}_{0\rightarrow 1}$ usually increases $\text{MisCover}_{1\rightarrow 0}$ and vice-versa. On Fashion-MNIST, we consider 6 ("shirt") vs. other classes. Only on CIFAR10, considering "vehicles" vs. "animals", mis-coverage cannot be reduced significantly. In particular, we were unable to reduce $\text{MisCover}_{1\rightarrow 0}$.

| $K_0 = 2$ vs. $K_1 =$ Others | | | | $K_0 =$ Even vs. $K_1 =$ Odd | | | |
|---|---|---|---|---|---|---|---|
| MNIST | | MisCover $\downarrow$ | | MNIST | | MisCover $\downarrow$ | |
| Method | Ineff | $0\rightarrow 1$ | $1\rightarrow 0$ | Method | Ineff | $0\rightarrow 1$ | $1\rightarrow 0$ |
| ConfTr | 2.11 | 49.68 | 14.74 | ConfTr | 2.11 | 38.84 | 38.69 |
| $L_{K_0,K_1}=1$ | 2.15 | 36.63 | 17.42 | $L_{K_0,K_1}=1$ | 2.16 | 29.36 | 49.08 |
| $L_{K_1,K_0}=1$ | 2.09 | 51.54 | 7.62 | $L_{K_1,K_0}=1$ | 2.09 | 44.3 | 26.08 |
| $K_0 = 6$ ("shirt") vs. $K_1 =$ Others | | | | $K_0 =$ "vehicles" vs. $K_1 =$ "animals" | | | |
| F-MNIST | | MisCover $\downarrow$ | | CIFAR10 | | MisCover $\downarrow$ | |
| Method | Ineff | $0\rightarrow 1$ | $1\rightarrow 0$ | Method | Ineff | $0\rightarrow 1$ | $1\rightarrow 0$ |
| ConfTr | 1.67 | 80.28 | 20.93 | ConfTr | 2.84 | 22.22 | 16.45 |
| $L_{K_0,K_1}=1$ | 1.70 | 72.58 | 25.81 | $L_{K_0,K_1}=1$ | 2.92 | 20.00 | 22.69 |
| $L_{K_1,K_0}=1$ | 1.72 | 81.18 | 17.66 | $L_{K_1,K_0}=1$ | 2.87 | 24.76 | 16.73 |

## N    MISCOVER RESULTS ON ADDITIONAL DATASETS

Tab. H provides mis-coverage results for different settings on MNIST, Fashion-MNIST and CIFAR10. As in the main paper, we are able to reduce mis-coverage significantly on MNIST and Fashion-MNIST. Only on CIFAR10, considering "vehicles" vs. "animals" as on CIFAR100 in the main paper, we are unable to obtain significant reductions. While, we are able to reduce $\text{MisCover}_{0\rightarrow 1}$ slightly from 22.22% to 20%, $\text{MisCover}_{1\rightarrow 0}$ increases slightly from 16.45% to 16.73% even for high off-diagonal weights used in $L$. Compared to CIFAR100, this might be due to less flexibility to find suitable trade-offs as CIFAR10 has only 10 classes. Moreover, mis-coverages on CIFAR10 are rather small to begin with, indicating that vehicles and animals do not overlap much by default.

## O    ADDITIONAL RESULTS ON BINARY DATASETS

Fig. F shows results complementing Fig. 5 (right) in the main paper. Specifically, we show that reducing inefficiency for class 1 ("good wine") is unfortunately not possible. This might also be due to the fact that class 1 is the majority class, with $\sim 63\%$ of examples. However, in addition to improving coverage conditioned on class 0, we are able to reduce coverage confusion $\Sigma_{0,1}$, $c.f.$ Sec. 3.3. We found that these results generalize to GermanCredit, however, being less pronounced, presumably because of significantly fewer training and calibration examples.

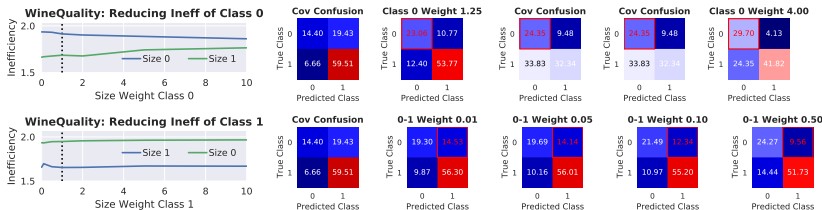

Figure F: **Manipulating Inefficiency and Coverage Confusion on WineQuality:** Complementing Fig. 5 (right) in the main paper, we plot the possible inefficiency reduction for class 1 ("good wine", left) and full coverage confusion matrices for increased $L_{0,0} > 1$ and $L_{1,0} > 0$ (right, top and bottom, respectively). While we can reduce inefficiency for class 0 ("bad wine"), this is not possible for class 1. However, class-conditional coverage for class 0 can be improved significantly and we can reduce coverage confusion $\Sigma_{0,1}$.

## P  PSEUDO CODE

Alg. B presents code in Python, using Jax (Bradbury et al., 2018), Haiku (Hennigan et al., 2020) and Optax (Hessel et al., 2020). We assume access to a smooth sorting routine that allows to compute quantiles in a differentiable way: `smooth_quantile`. Specifically, Alg. B provides an exemplary implementation of ConfTr with (smooth) THR and $\mathcal{L}_{\text{class}}$ as outlined in Alg. 1 in the main paper. `smooth_predict_threshold` and `smooth_calibrate_threshold` implement differentiable prediction and calibration steps for THR. These implementations are used in `compute_loss_and_error` to "simulate" CP on mini-batches during training. Size loss $\Omega$ from Eq. (3) and classification loss from Eq. (5) are implemented in `compute_size_loss` and `compute_general_classification_loss`. Note that the definition of `compute_loss_and_error` distinguishes between `trainable_params` and `fixed_params`, allowing to fine-tune a pre-trained model.

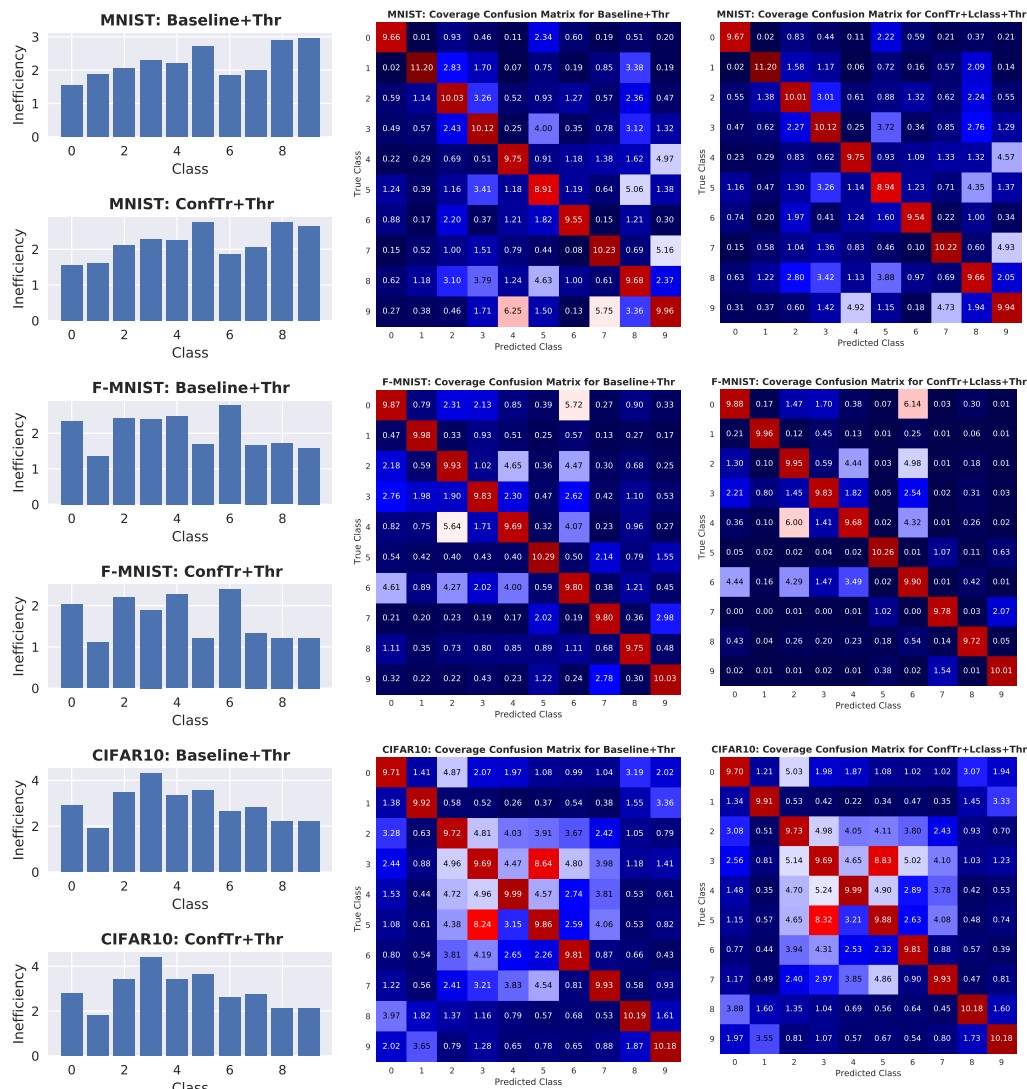

Figure G: **Class-Conditional Inefficiency and Coverage Confusion:** Comparison between baseline and ConfTr regarding class-conditional inefficiency and coverage confusion $\Sigma$, *c.f.* Sec. 3.3. For the inefficiency comparison, we consider ConfTr *without* $\mathcal{L}_{\text{class}}$, while for coverage confusion, ConfTr was trained with $\mathcal{L}_{\text{class}}$. As ConfTr reduces overall inefficiency quite significantly on MNIST and Fashion-MNIST, class-conditional inefficiency is also lower, on average. But the distribution across classes remains similar. The same holds for coverage confusion, where lower overall inefficiency reduces confusion across the matrix, but the "pattern" remains roughly the same. On CIFAR10, ConfTr does not improve average inefficiency significantly, such that the confusion matrix remains mostly the same.

**Algorithm B Python Pseudo-Code for ConfTr:** We present code based on our Python and Jax implementation of ConfTr. In particular, we include smooth calibration and prediction steps for THR as well as the classification loss $\mathcal{L}_{\text{class}}$ and the size loss $\Omega$. Instead of including a full training loop, `compute_loss_and_error` shows how to compute the loss which can then be called using `jax.value_and_grad(compute_loss_and_error, has_aux=`**`True`**`)` and used for training using Optax. Hyper-parameters, including `alpha`, `dispersion`, `size_weight`, `temperature`, `loss_matrix`, `size_weights` and `weight_decay`, are not defined explicitly for brevity. `smooth_quantile` is assumed to be a provided differentiable quantile computation method. Finally, `model` can be any Jax/Haiku model.

```python
import jax
import jax.numpy as jnp
import haiku as hk

def smooth_predict_threshold(
    probabilities: jnp.ndarray, tau: float, temperature: float) -> jnp.ndarray:
  """Smooth implementation of prediction step for Thr."""
  return jax.nn.sigmoid((probabilities - tau) / temperature)

def smooth_calibrate_threshold(
    probabilities: jnp.ndarray, labels: jnp.ndarray,
    alpha: float, dispersion: float) -> float:
  """Smooth implementation of the calibration step for Thr."""
  conformity_scores = probabilities[jnp.arange(probabilities.shape[0]), labels.astype(int)]
  return smooth_quantile(array, dispersion, (1 + 1./array.shape[0]) * alpha)

def compute_general_classification_loss(
    confidence_sets: jnp.ndarray, labels: jnp.ndarray,
    loss_matrix: jnp.ndarray) -> jnp.ndarray:
  """Compute the classification loss Lclass on the given confidence sets."""
  one_hot_labels = jax.nn.one_hot(labels, confidence_sets.shape[1])
  l1 = (1 - confidence_sets) * one_hot_labels * loss_matrix[labels]
  l2 = confidence_sets * (1 - one_hot_labels) * loss_matrix[labels]
  loss = jnp.sum(jnp.maximum(l1 + l2, jnp.zeros_like(l1)), axis=1)
  return jnp.mean(loss)

def compute_size_loss(
    confidence_sets: jnp.ndarray, target_size: int, weights: jnp.ndarray) -> jnp.ndarray:
  """Compute size loss."""
  return jnp.mean(weights * jnp.maximum(jnp.sum(confidence_sets, axis=1) - target_size, 0))

FlatMapping = Union[hk.Params, hk.State]
def compute_loss_and_error(
    trainable_params: FlatMapping, fixed_params: FlatMapping, inputs: jnp.ndarray,
    labels: jnp.ndarray, model_state: FlatMapping, training: bool, rng: jnp.ndarray,
) -> Tuple[jnp.ndarray, FlatMapping]:
  """Compute classification and size loss through calibration/prediction."""
  params = hk.data_structures.merge(trainable_params, fixed_params)
  # Model is a Haiku model, e.g., ResNet or MLP.
  logits, new_model_state = model.apply(params, model_state, rng, inputs, training=training)
  probabilities = jax.nn.softmax(logits, axis=1)

  val_split = int(0.5 * probabilities.shape[0])
  val_probabilities = probabilities[:val_split]
  val_labels = labels[:val_split]
  test_probabilities = probabilities[val_split:]
  test_labels = labels[val_split:]
  # Calibrate on the calibration probabilities with ground truth labels:
  val_tau = smooth_calibrate_threshold(val_probabilities, val_labels, alpha, dispersion)

  # Predict on the test probabilities:
  test_confidence_sets = smooth_predict_threshold(test_probabilities, val_tau, rng)
  # Compute the classification loss Lclass with a fixed loss matrix L:
  classification_loss = compute_general_classification_loss(
      test_confidence_sets, test_labels, loss_matrix)
  # Optionally set size weights determined by ground truth labels:
  weights = size_weights[test_labels]
  # Compute size loss multiplied by size weight:
  size_loss = size_weight * compute_size_loss(confidence_sets, weights)

  # Compute the log of classification and size loss:
  loss = jnp.log(classification_loss + size_loss + 1e-8)
  loss += weight_decay * sum(jnp.sum(jnp.square(param)) for param in jax.tree_leaves(params))

  return loss, new_model_state
```

