# OpenReview forum: "Learning Optimal Conformal Classifiers"
_ICLR.cc/2022/Conference — ICLR 2022 Spotlight_

### Official Review · Reviewer_uxwS · 2021-11-02

**Correctness:** 2
**Technical Novelty And Significance:** 3
**Empirical Novelty And Significance:** 3
**Recommendation:** 8
**Confidence:** 3

**Main Review:**

I think the paper intent is very interesting, as it would clearly be interesting to learn optimal conformal predictors rather than plugging them in once the model is learned, possibly using a loss function not entirely correlated to the purpose of providing set-valued calibrated predictions.

I cannot say that I was able to fully follow the development of the proposed method, as it is densely presented in 2 pages. I would have appreciated some more elaborated examples or illustration of how the derived loss function and subsequent techniques approximate the whole conformal prediction framework, but even the (many) appendices did not provide additional intuition for the less expert read, and focused mostly on providing additional experimental results. In terms of accessibility to a larger audience, this is clearly a weakness that could be resolved by being more pedagogical (possibly in the appendices).

This said, I still have a couple of questions about the developed method:

* A first one is that it is claimed that the proposed approach retains the conformal guarantees. However, I am not sure I fully understand why this should be the case theoretically, as what is used in the paper are differentiable approximations of the classical approaches. If those are approximations, why should those retain the same theoretical properties of calibration as the initial procedure? What guarantees, theoretically, that we will not be too far from the calibrated results? If there is no such guarantees, then at least some empirical calibration lines (in the appendices?) should be provided to see if indeed validity is still reasonably preserved.

* A second one is that it is often claimed in the paper that conformal prediction is unable to handle class-wise issues, which is partially untrue  as such problems can be handled, for instance, by using so-called Mondrian conformal predictors over various categories, that can be user-defined and is typically used to have class-wise conformal guarantees. Could it be explained how the presented framework would handle such an issue? Indeed, most of the discussion about application-specific losses focus on efficiency and the shape of the conformal prediction sets, but not how one could control class-wise errors (the task that Mondrian predictors can handle quite well)

* While I think that the proposal to better control the shape of the predicted set is actually quite interesting, I would be interested in having a more detailed argument about why this would preserve conformal guarantees (in the same way that lossless formulation would)? I am also missing a more critical discussion about the risks of privileging higher accuracy on some instances over a lower accuracy on some potentially critical instances? Classical marginal conformal predictors are known to have such undesirable effects, but it is unclear to me whether the proposed formulation can control in any sense this kind of behaviour? Would not an increased efficiency for a class/case lead to a lower accuracy for this class, possibly counter-balanced by a higher accuracy for the class for which we allow the learning method to be more uncertain about? I believe this would at least deserve a refined calibration/validity study along with the analysis of result efficiency.

* Small comment: In the abstract and beginning of introduction, I would not especially refer to posterior probability estimates, as conformal predictions merely need score-valued predictors (whether those are brought back to the interval [0,1] by a softmax or another transformation does not matter much). I would also avoid talking about Bayes decision rule being the most probable class, as this is only true if one consider a zero/one loss?

**Summary Of The Paper:**

Conformal prediction is a method that intends to provide set-valued, calibrated prediction, in the sense that they cover the true class with a guaranteed marginal statistical accuracy. The paper mainly proposes to learn the conformal predictor at the same time as the predictive model is used, through an end-to-end procedure.

To this effect, conformity scores and threshold values (using quantiles) are approximated by differentiable surrogates that are directly injected into the learning framework. It is also argued that the framework can offer some solutions to better control class-wise predictions, or class-wise constraints on the conformal sets.

**Summary Of The Review:**

The paper proposes a differentiable scheme to learn conformal predictors. This is clearly an interesting approach that could lead to improve conformal predictors, which are gaining increased attention in ML (but existing as a consolidated approach since about 15 years). The technical part of the paper is densely presented, making it difficult to really follow. It is also unclear to which extent the validity guarantees coming with plug-in conformal approaches are actually preserved. However, although no theoretical guarantee of that is fully discussed, the differentiable approximation seems close enough to preserve at least a reasonable empirical validity. The additional loss-based formulation and its various adaptations look quite interesting, but it is here even less obvious what are their relation to class-wise conformal methods (i.e., Mondrian ones), and what are the impacts of choosing different losses over validities (particularly class-wise ones).

---

> ### Author Response · Authors · 2021-11-17
> **Class-Conditional Conformal Prediction and Trade-Offs**
>
> We thank the reviewer for the detailed review and address questions one-by-one. The point raised regarding the coverage guarantee is addressed in a general comment above.
>
> **Clarity and writing**: We appreciate the honest feedback and acknowledge that our paper is dense. We will try to include more discussion and reduce details as appropriate. We hope that our clarifications regarding the obtained coverage guarantee in the general comment above will already contribute to a better understanding of our paper.
>
> **Class-conditional conformal prediction**: In the paper, we mostly focus on inefficiency and the composition of confidence sets. This is because, as the reviewer noted, class-conditional coverage is possible using special conformal predictors, including the class-conditional one by Sadinle et al. (2019). As our conformal training is entirely complementary to the conformal predictor used at test time, class-conditional coverage can be guaranteed simply by employing a class-conditional conformal predictor after training with our approach, including the Mondrian conformal predictors mentioned by the reviewer. Besides this straightforward approach, we could (a) empirically improve class-conditional coverage by putting higher weight on classes with low empirical coverage in the proposed classification loss or (b) train directly with a smooth, class-conditional conformal predictor (for the one by Sadinle et al. (2019) this would be a simple extension of the one presented in the paper).
>
> **Coverage guarantee and trade-offs**: We agree with the reviewer that reduced inefficiency  for specific classes might also lead to poorer coverage on these classes. While we did not investigate this phenomenon in detail, we expect there to be a trade-off similar to the one when addressing coverage confusion. However, this could be tackled using a class-conditional conformal predictor at test (and potentially training) time as described above. While this generally increases inefficiency across all classes, especially harder ones, our conformal training could still try to explicitly minimize inefficiency of specific classes without losing the class-conditional coverage guarantee.
>
> **Abstract and introduction**: We appreciate these comments on writing and will change this accordingly. We know that conformal prediction is a very general framework and mostly independent of the specifics of how the scores are obtained. However, in order to bring training and conformal prediction closer together, we decided to explicitly consider the posterior probabilities as these are commonly dealt with in the context of deep learning.

---

> > ### Comment · Reviewer_uxwS · 2021-11-20
> > **Thank you for clarifications**
> >
> > I thank the authors for their answers to my concerns, which mainly concerned coverage guarantees.

---

### Official Review · Reviewer_8BvY · 2021-11-02

**Correctness:** 3
**Technical Novelty And Significance:** 3
**Empirical Novelty And Significance:** 3
**Recommendation:** 8
**Confidence:** 3

**Main Review:**

Strengths

1. The proposed method addresses an important problem and the associated technical area has seen a surge in publications including a workshop at ICML 2021.

2. The proposed method combines several novel elements and appears to be theoretically sound.

3. The proposed method is shown to outperform a competing method by Belloti in terms of the inefficiency measure (for the same coverage guarantee) on 5 public multi-class datasets (including CIFAR100) and a binary dataset (wineQuality).
The authors attribute this to the fact that Belloti's method does not incorporate the calibration step within the overall training process.

4. The authors convincingly demonstrate how the size loss term's weight can control the class-conditional inefficiency in Fig. 3 and how the "configurable" classification loss can control coverage confusion and mis-coverage in Fig. 5

5. The thorough appendices cover important aspects as impact of hyperparameters, random trials, python source, etc.

Weaknesses

The weaknesses are relatively minor and mainly related to clarity or typos:

1.  In Table 1, the authors report additional inefficiency improvements over ConfTR with the configurable L_{class}, but they don't seem to provide the configuration parameters for L_{class}.

2. In the text below Table 1, the authors mention: "we found that Bel or ConfTr do not necessarily recover the accuracy of the baseline". It would be nice if the authors report the actual accuracies in a table (with the specified value of \alpha) and also define exactly how the accuracy given a predictive set was computed (presumably only singleton sets could count towards accurate predictions).

3. Mis-spelling: "CIRFAR10" in Table 2

4. Grammar / wording in a sentence in Sec. 4.2 can be improved: ... the more difficult class 3 (“cat”) obtains higher inefficiency than to the easier class ...


**Summary Of The Paper:**

Conformal prediction methods generally operate via post-processing on black-box classifiers to produce a confidence set. In this paper, the authors integrate this post-processing step into the training procedure in order to reduce the inefficiency of conformal prediction while providing the same coverage guarantee.
The proposed method uses soft thresholding and differentiable sorting to convert two popular conformal post-processing method classess (THR - Sadinle et al. and APS - Romano et al.) into differentiable modules, although they only use THRLP (the method from Sadinle et al. applied to log-probabilities) in the training process for the results (in Table 1).  Additionally, they propose the use of weighted inefficiency regularization (also termed the size loss) and a "configurable" classification loss that can yield desirable behavior w.r.t. class-conditional inefficiency and mis-coverage (or coverage confusion) respectively.
The proposed method is shown to outperform a competing method by Belloti on 5 public multi-class datasets (including CIFAR100) and a binary dataset (wineQuality).

**Summary Of The Review:**

The authors propose a novel, impactful and theoretically sound method for conformal training and convincingly demonstrate its efficacy.
This reviewer has some minor concerns regarding clarity that can be readily addressed. Therefore, I recommend acceptance.

---

> ### Author Response · Authors · 2021-11-17
> **Clarification of L_class**
>
> We thank the reviewer for the comments and address them in the following. The second point regarding accuracy and coverage in Table 1 is addressed in a common comment above.
>
> **Clarification regarding L_class**: The loss matrix L in L_class used in Table 1 for inefficiency improvements is the identity matrix, $L = I_K$ where K is the number of classes. This means that L_class enforces the true class to be included in the confidence set, but does not add any additional constraints (as the off-diagonal of L is zero). For the coverage confusion experiments, in contrast, non-zero off-diagonals are used in addition to the 1s on the diagonal. We will make the choice for L explicit in the paper.

---

> > ### Comment · Reviewer_8BvY · 2021-11-30
> > **Thank you for the revisions**
> >
> > Thank you for the clarifications on both my questions and for making the corresponding changes to the paper. I am also happy with the revisions addressing other reviewer concerns.

---

### Official Review · Reviewer_1Awq · 2021-11-03

**Correctness:** 3
**Technical Novelty And Significance:** 2
**Empirical Novelty And Significance:** 2
**Recommendation:** 5
**Confidence:** 3

**Main Review:**

**Strong points:**
* The idea of putting conformal inference into the neural network training loop is interesting.
* The goal of reducing confidence set size is important.

**Weak points:**
* The improvements (e.g., in Tables 1,2) seem somewhat minor to me.  Am I missing something?
* It's actually not totally clear to me that you get coverage at the nominal level -- can you justify that?
* I don't really see what the proposal here offers over Bates et al. (2021) -- can you explain?
* I found the paper somewhat hard to read -- I think things could have been explained a bit better in places.  But also notation was often used without defining it first, and some non-standard terminology was used.

**Questions:** see my "Weak points" and "Additional feedback" sections.

**Additional feedback:**
* How do the results change as you vary the mini-batch size?
* Eq 3 -- I think you forgot to put $|\cdot|$ around the confidence sets inside the max?
* You appear to use $E_\theta$ before defining it.

**Summary Of The Paper:**

**Summary:**
* The paper considers putting conformal inference into the process of fitting deep neural networks, i.e., the paper considers "differentiable conformal prediction" / back-prop'ing through conformal inference.
* The goal is to reduce the size of the prediction sets generated by conformal inference -- and more specifically to reduce the size variably in different parts of the input space.
* Some experiments are given to show that the confidence set sizes are indeed smaller w/ the proposal than w/o it.


**Summary Of The Review:**

**Recommendation:** reject.  I have some questions about the novelty of the empirical results and the methodology.  Happy to change my mind if I missed something.

---

> ### Author Response · Authors · 2021-11-17
> **Inefficiency Improvements and Difference to (Bates et al., 2021)**
>
> We appreciate the reviewer’s feedback and address the raised points one-by-one. Note that the point on the obtained coverage guarantee is covered in a common comment above. We hope that clarifying the coverage guarantee and the complementarity of our approach to Bates et al. (2021) convinces the reviewer to increase the score.
>
> **Inefficiency improvements**: Regarding the inefficiency results in Table 1, we need to distinguish between the two conformal predictors used:
>
> Regarding APS of Romano et al. (2020), we emphasize that the obtained inefficiency improvements are significant: They are lowest on MNIST (14.4% improvement) and CIFAR10 (11.1%) but significantly higher with more classes on EMNIST (52 classes,  32.4%) and CIFAR100 (100 classes, 23.4%). From Figure 2 (right), we outperform the recently published regularized version of APS by Angelopoulos et al. (2021).
>
> Regarding Thr by Sadinle et al. (2019), absolute improvements are smaller. This is because Thr is, given a fixed model, a very efficient conformal predictor. Nevertheless, we can obtain good relative improvements on some datasets, e.g., 18% on Fashion-MNIST or 13% on EMNIST, even for smaller $\alpha = 0.001$. On CIFAR10 and CIFAR100, with 3.1 and 1.8%, respectively, improvements are smaller also because we use pre-trained features (obtained using cross-entropy training).
>
> Overall, we believe that improvements in between 1.8% (for Thr on CIFAR100) and 32.4% (for APS on EMNIST) can be significant in many practical applications and we provide a proof-of-concept on standard benchmark datasets. Similarly, the results in Table 2 (left) are meant as a proof-of-concept for using conformal training for conformalizing ensembles. We will stress the relative inefficiency improvements in the main paper more clearly.
>
> **Difference to Bates et al. (2021)**: We agree with the reviewer that our motivation of tackling application specific objectives is similar to Bates et al. (2021). In fact, because our conformal training is just a novel training procedure, it is easily possible to apply Bates et al. (2021) after training. In this case, it is even possible to use the same or similar objectives during training and for conformalization (providing a guarantee on this objective). Moreover, conformal training could also be extended by employing Bates et al. (2021) during training. Similar to the smooth conformal predictors presented in the papers, this would require a smooth version of Bates et al. (2021). This highlights the flexibility of our approach and emphasizes the complementarity to advancements in conformal prediction.
>
> **Writing and notation**: We agree with the reviewer that the paper is dense. We will specifically revisit notation and terminology but would appreciate any specific pointers by the reviewer which parts were hard to follow and could be improved.

---

> > ### Comment · Reviewer_1Awq · 2021-11-29
> > **Response to the response**
> >
> > First of all, thanks for the additional work, and for spending the time to write up a response!
> >
> > * Re: your point on the "[c]overage guarantee for conformal training" -- I'm a little confused here.  Are you trying to say that: (i) your conformal training methodology itself doesn't have a coverage guarantee, but (ii) if you performed conformal inference **as usual** after conformal training, then you would (of course) have a coverage guarantee?  If so, I don't really see how that point is very ... useful/interesting/etc., as it is essentially orthogonal to your entire paper (i.e., it applies to **any** base model, not just yours)?
> >
> > * Re: your point on "[i]nefficiency improvements" -- on Table 1, I get that the percentage improvement (14.4%) is large, but in raw terms, you are reducing the average set size from 2.50 (attained by "Basel.") $\rightarrow$ 2.14 (attained by "$\mathcal L_{\textrm{class}}$") on MNIST.  On CIFAR-100, you are reducing the average set size from 16.62 $\rightarrow$ 12.73.  Given that there are 100 classes in CIFAR-100, and 10 classes in MNIST, aren't these kind of small improvements (in raw terms)?  Essentially, you are excluding **at most** 1 label on average from your sets on MNIST (which seems not so useful to me), and roughly 4 on CIFAR-100 (which is good) ... what am I missing here?
> >
> > * The results on Table 2 seem a bit better, though I do not understand why $\alpha$ is so tiny there (e.g., $\alpha = 0.001$ is unusually small) ...

---

> > > ### Author Response · Authors · 2021-11-29
> > > **Clarification**
> > >
> > > Thanks for getting back to us for clarification.
> > >
> > > Regarding the coverage guarantee: This is indeed how it works for conformal training. The obtained model can be handled as any other base model, as you mentioned. And the orthogonality you mention is actually one of the key advantages of conformal training: At test time, you can still use your favorite conformal predictor with the corresponding guarantee(s). However, by using conformal training during training you can improve key metrics such as inefficiency compared to "standard" training (which means cross-entropy training in our paper). This means that conformal training is entirely complementary to future work on conformal predictors or conformity scores.
> > >
> > > On the inefficiency improvements: You are not missing anything here. However, we believe that these improvements are in fact significant - not only in relative but also in absolute terms.
> > >
> > > On CIFAR100, for example, we believe a reduction of inefficiency by 4 to be very significant. Thinking about dermatological condition classification, usually with hundreds of possible conditions (Liu et al., 2020; Roy et al., 2021; Jain et al., 2021), predicting 4 fewer conditions on average also means that docotrs have to consider less conditions, conduct less tests and invest less time. For patients this could, in the extreme case, make the difference between certainty about their condition (confidence set of size 1), and high uncertainty (confidence set of size 5) resulting in unnecessary anxiety - subject to the coverage guarantee of 99%. In such settings, a reduction in inefficiency automatically translates to better patient care and saved resources. We believe that even a reduction from 2.5 to 2.14 on MNIST can be important in many settings. For example, it also means that more examples can be classified without uncertainty (confidence set of size 1). Even if these are only few examples (relatively speaking), in large-scale applications it can be a advantage that decisions on these examples do not need human correction/intervention or other redundancies built-in.
> > >
> > > We also want to emphasize that similar improvements are viewed as significant in related work, e.g., for RAPS by Angelopoulos et al. (2021) which we outperform. Moreover, while the threshold conformal predictor by Sadinle et al. (2019) is argued to be optimal in terms of inefficiency (for a fixed model), training using our proposed conformal training can further reduce inefficiency compared to standard training. We think this further supports our motivation that conformal prediction needs to be considered during model training.
> > >
> > > Regarding the low confidence level $\alpha$, we believe that very small confidence levels - even though not frequently considered in related work - will become very important in many high-stakes and security critical domains. However, in the paper, this experiment was mainly meant as proof-of-concept that the improvements obtained via conformal training with, e.g., $\alpha = 0.01$ during training also generalize to lower $\alpha < 0.01$ without requiring re-training.

---

### Official Review · Reviewer_iXzv · 2021-11-03

**Correctness:** 4
**Technical Novelty And Significance:** 3
**Empirical Novelty And Significance:** Not applicable
**Recommendation:** 8
**Confidence:** 4

**Main Review:**

## Strengths

The paper builds a convincing case for the proposed method by calling attention to the mismatch of goals in traditional training procedures and conformal prediction. Once one accepts this motivation, the proposed method appears quite reasonable. The experiments appear to be quite thorough. The extension to other structure-inducing losses is interesting, although its actual deployment may warrant caution.

## Weaknesses

I found no significant weaknesses in my initial assessment.

A few questions for consideration are as follows:

1. How large should the size of each mini-batch be? This probably depends on $\alpha$, but are there other things that should be taken into account? Also, I feel like Section G ought to appear in the paper.

2. In **Conformal Predictors for Training**, it is reported that "Bel or ConfTr do not necessarily recover the accuracy of the baseline. When training from scratch, accuracy can be 2--6% lower while still *reducing* inefficiency." Here, I understand "accuracy" to mean the accuracy of marginal coverage. In that case, why is there any loss of accuracy at all when all methods are calibrated on a separate held-out set?

## Typos
- (p. 3) on test example -> on a test example
- (p. 6) Put parentheses around "see Tab. 1 for the main results"
- (p. 9. The last line) loose -> lose
- (p. 10. The 2nd paragraph) loosing -> losing
- (p. 14. Section B) Remove the parentheses around "Vovk, 2021; Barber et al., 2019b"
- (p. 17. Table B) Batch Size and Learning Rate switched in the row corresponding to "Camelyon, ConfTr+$\mathcal{L}_{\text{class}}$"

**Summary Of The Paper:**

In split conformal prediction, conformalization is applied using a model that has been trained on a separate training set. However, the training procedure usually optimizes an objective that has little to do with the ultimate goal of producing informative prediction sets. For the problem of classification, the paper proposes an alternative training procedure that simulates conformalization on mini-batches so that the learned model is directly optimized to produce smaller prediction sets. In addition, the proposed alternative training procedure can accommodate other loss functions, which can induce certain structures in the prediction sets. The performance gains are demonstrated through experiments.

**Summary Of The Review:**

This is a well-argued paper backed up by extensive experimental results. To my knowledge, the method represents a novel contribution. I would recommend it for acceptance.

---

> ### Author Response · Authors · 2021-11-17
> **Batch Size During Conformal Training**
>
> We appreciate the reviewer’s feedback and briefly address the raised point about batch size during training. The question regarding accuracy of conformal training is discussed in the common comment above.
>
> **Batch size for conformal training**: As mentioned by the reviewer, the main concern when determining batch size is indeed the used confidence level $\alpha$. Other aspects did not impact conformal training significantly. Table D in Appendix G highlights that, for example, a batch size of 100 might not be sufficient, since only half of the batch (50 examples) is used for calibration which is not sufficient for $\alpha = 0.01$. This is because smooth calibration gets too sensitive to outliers, resulting in large variations of empirical overage on the other half of the batch. This is confirmed on Table G, where we have to use $\alpha = 0.05$ for allowing smaller batch sizes of 20 or 10 on Camelyon. Beyond that, batch size is chosen as usual in deep learning. For example, a batch size of 1000 performs slightly worse, which might be due to poorer training dynamics, as commonly observed in large batch training. We will consider discussing this in the main paper.

---

> > ### Comment · Reviewer_iXzv · 2021-11-30
> > **Batch size**
> >
> > Thank you for the clarification. I still feel like it would be helpful to move some of the discussions in the appendix about the batch size selection, as this is a hyperparameter that is explicitly introduced by the proposed method, but I am also okay with it happening in the appendix. Overall, I see no reason to change my score in either direction.

---

### Author Response · Authors · 2021-11-17
**General Comment Clarifying Coverage Guarantee and Accuracy Metric**

We appreciate the reviewers’ thorough and constructive feedback. All reviewers agree that our idea of training classifier and conformal predictor end-to-end is interesting and novel.

The following two points were raised by multiple reviewers:

**Coverage guarantee for conformal training**: We acknowledge that the coverage guarantee obtained by our proposed conformal training procedure is not stated explicitly. This important point will be clarified in the revised version of the paper. Essentially, conformal training is a novel training procedure that simulates conformalization on each mini-batch, independent of the conformal predictor used after training. While we introduce smooth versions of recent conformal predictors to accomplish this, these are only used during training. After training, i.e., at test time, the model is re-calibrated (on a held-out calibration set disjoint from the training set) using any existing conformal prediction method. Thus, the coverage guarantee we obtain is a direct result of the conformal predictor used at test time, e.g., Sadinle et al. (2019) or Romano et al. (2020). In essence, because we only propose a different way of training the model, coverage at test time is guaranteed by construction of the conformal predictor used after training. We empirically validated this, but did not report these results for brevity.

Being independent of the conformal predictor used at test time makes conformal training a very flexible training procedure: We do not need to train and test with the same conformal predictor. This makes conformal training entirely complementary to existing and future work on conformal prediction. For example, as mentioned by some reviewers, we can readily apply the approach by Bates et al. (2021) or any class-conditional conformal predictor, e.g., Sadinle et al. (2019), after conformal training and obtain the corresponding guarantees.

The smooth conformal predictors used during training do not necessarily provide coverage guarantees. This is because our smooth implementations are not exact, as is commonly the case when integrating non-smooth operations in deep learning. Nevertheless, during training, we obtain empirical coverage close to $(1 - \alpha)$. This is because, for temperature T and dispersion $\epsilon$ approaching 0, the non-smooth counter-parts are recovered. This is sufficient to make conformal training work well. However, we agree that there might be more principled approaches for obtaining smooth conformal predictors with guaranteed coverage.

We will make sure to clarify the above points in the final version of the paper.

**Accuracy and marginal coverage**: We agree with the reviewers that our statement on not recovering the baseline accuracy using conformal training may lead to confusion and will clarify this statement in the revised paper. First, we want to clarify the nomenclature of the used metrics: With “accuracy” we refer to the top-1 accuracy as evaluated using the predicted posterior probabilities, i.e., whether the argmax predictions are the true classes. This means that accuracy is not evaluated on the confidence sets obtained through conformal prediction and does not correspond to the coverage on singleton confidence sets. “Coverage”, in contrast, evaluates whether the true classes are included in the confidence sets. As discussed above, marginal coverage is guaranteed. Table E and F show that the accuracy for conformal training can be lower compared to the baseline (trained using cross-entropy loss). We highlight this in the main paper because we find it surprising that conformal training does not recover the baseline accuracy but still improves inefficiency. In our opinion, this highlights that training with cross-entropy loss is not a good surrogate when aiming to reduce inefficiency, which is the motivation of our paper.

---

### Author Response · Authors · 2021-11-22
**Initial Revision of Paper**

We also uploaded an initial draft of the revised version, taking into account many of the reviewers’ suggestions. All changes are marked in red, major changes are:
* Throughout the paper, we clarified the use of conformal training as training procedure only, while any conformal predictor can be applied after training to obtain the corresponding coverage guarantee.
* In Section 2.2, the discussion of smooth APS (Romano et al., 2020) has been moved to Appendix D to make space for a discussion of the coverage guarantees of the smooth threshold conformal predictor following Sadinle et al. (2019) as well as the guarantees obtained after training.
* We explicitly highlight our definition of accuracy.
* Table 1 now includes percentage improvements.

---

### Decision · Program_Chairs · 2022-01-20

**Decision:**

Accept (Spotlight)

**Comment:**

In this paper, a new learning scheme for minimizing the confidence set by conformal prediction is proposed. Most of the reviewers agree that the idea is interesting and novel. This is an important contribution to trustworthy ML, with theoretically sound considerations and thorough experimental validation.